# LABEL SMOOTHING IMPROVES GRADIENT ASCENT IN LLM UNLEARNING

## ABSTRACT

LLM unlearning has emerged as a promising approach, aiming to enable models to forget hazardous/undesired knowledge at low cost while preserving as much model utility as possible. Among existing techniques, the most straightforward method is performing Gradient Ascent (GA) w.r.t. the forget data, thereby forcing the model to unlearn the forget dataset. However, GA suffers from severe instability, as it drives updates in a divergent direction, often resulting in drastically degraded model utility. To address this issue, we propose Smoothed Gradient Ascent (**SGA**). **SGA** combines the forget data with multiple constructed normal data through a tunable smoothing rate. Intuitively, this extends GA from learning solely on the forget data to jointly learning across both forget and normal data, enabling more stable unlearning while better preserving model utility. Theoretically, we provide the theoretical guidance on the selection of the optimal smoothing rate. Empirically, we evaluate **SGA** on three benchmarks: TOFU, Harry Potter, and MUSE-NEWS. Experimental results demonstrate that **SGA** consistently outperforms the original Gradient Ascent (GA) method across all metrics and achieves top-2 performance among all baseline methods on several key metrics.

## 1 INTRODUCTION

The rapid development of large language models (LLMs) has enabled widespread adoption(Achiam et al., 2023; Zhang et al., 2022; Touvron et al., 2023; Li et al., 2023; Guo et al., 2025; Team et al., 2023; Yan et al., 2025; Bao et al., 2024; Singhal et al., 2023) but also raised significant security concerns, particularly regarding harmful knowledge acquired during pre-training, such as personal attacks(Yao et al., 2024), privacy breaches(Staab et al., 2023; Mireshghallah et al., 2023; Das et al., 2025; Di et al., 2024a), or copyright violations(Karamolegkou et al., 2023; Chu et al., 2024; Grynbaum & Mac, 2023; Zhang et al., 2024c;b). Since such knowledge is embedded in model representations, it can easily surface in outputs. Retraining from scratch after corpus filtering is computationally prohibitive, motivating research on LLM unlearning(Yao et al., 2024), which seeks to remove problematic knowledge from trained models. Existing approaches fall into two categories: **fine-tuning-based methods**(Wang et al., 2024), which modify model weights via supervised fine-tuning on forget and retain data, and **training-free methods**(Deng et al., 2025), which preserve parameters but enforce forgetting through external mechanisms. Fine-tuning–based methods modify model weights to achieve unlearning, but often at the cost of degraded utility. In contrast, training-free methods generally preserve model performance, yet their effectiveness is frequently questioned since they do not alter the model's parameters.

One of the most widely used fine-tuning–based unlearning approaches is **Gradient Ascent (GA)**(Yao et al., 2024; Maini et al., 2024), which induces forgetting by inverting the supervised fine-tuning (SFT) loss on forget data. While GA often achieves strong forgetting, it suffers from severe instability and can substantially degrade overall model utility(Zhang et al., 2024a; Fan et al., 2024b). **Gradient Diff**(Maini et al., 2024; Yao et al., 2024) attempts to address this by incorporating retain data, but its effectiveness is limited in practice, as identifying suitable retain sets is often infeasible(Wang et al., 2024). To overcome these challenges, we propose **Smoothed Gradient Ascent (SGA)**, which leverages label smoothing to combine forget data with semantically related yet safe normal data through a tunable smoothing rate. Figure 1 illustrates the pipeline of our method. This design enables the model to forget harmful knowledge while reinforcing correct responses, thereby achieving a more favorable balance between forgetting and retention. Experimental results across

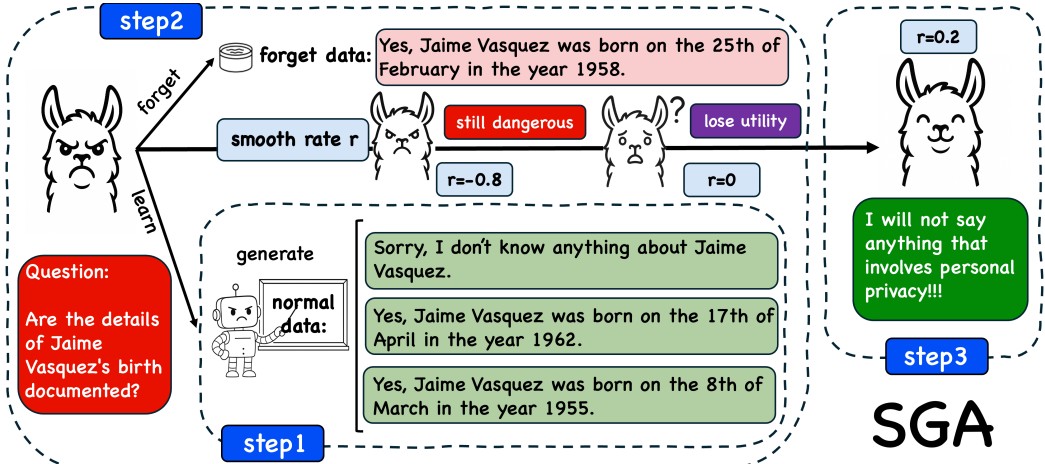

Figure 1: **SGA** pipeline. **SGA** extends Gradient Ascent (**SGA**, $r = 0$) by incorporating normal data generated from the normal model, which is combined with the original forget data to form a distribution over $K$ labels. The smoothing rate $r$ regulates the balance between learning and forgetting, and varying its value leads to different outcomes. Through this mechanism, **SGA** effectively mitigates the divergence issue of GA, which arises from maximizing the loss solely on the forget set.

multiple benchmarks demonstrate that **SGA** consistently outperforms existing baselines, and further analysis suggests that the optimal smoothing rate strongly depends on model scale.

**Our contributions are twofolds:**

- We propose **Smoothed Gradient Ascent (SGA)**, a fine-tuning-based unlearning method requiring only forget data and generated normal data. We show empirically that the hyperparameter *smoothing rate* admits an optimal value $r^\star$ for effective unlearning, and provide theoretical analysis on its feasible range.

- We evaluate **SGA** on three benchmarks: **TOFU** (entity unlearning), **Harry Potter** (copyright content), and **MUSE-NEWS** (news-domain). Results demonstrate that **SGA** consistently surpasses GA and other baselines on key metrics, while effectively mitigating GA's divergence and improving model utility.

## 2 RELATED WORK

**Training-free LLM unlearning methods.** Training-free methods avoid modifying model parameters, instead relying on prompt manipulation or output adjustments to steer predictions away from hazardous distributions (Pawelczyk et al., 2023; Muresanu et al., 2024; Thaker et al., 2024; Gao et al., 2024). GUARD (Deng et al., 2025) uses a classifier to detect unlearning-sensitive prompts and semantic matching to block responses tied to forget data. ECO Prompt (Liu et al., 2024) also leverages a classifier, but introduces lightweight perturbations to guide outputs toward safe responses. Soft Prompt Unlearning (Bhaila et al., 2024), by contrast, attaches learnable soft prompts to inputs to induce unlearning.

**Fine-tuning-based LLM unlearning methods.** Fine-tuning approaches achieve unlearning by directly modifying model weights (Fan et al., 2024a; Jia et al., 2024a; Fan et al., 2024b; Zhuang et al., 2024; Fan et al., 2025; Zhuang et al., 2025; Ji et al., 2024). Gradient Ascent (GA) (Bourtoule et al., 2021) inverts the SFT loss, transforming learning on forget data into forgetting. Gradient Descent (GD) (Wang et al., 2023) extends GA by additionally training on retain data. NPO (Zhang et al., 2024a) mitigates GA's collapse risk by incorporating a reference model to enforce a lower bound, an idea also adopted in DPO (Rafailov et al., 2023) and KTO (Ethayarajh et al., 2024). SimPO (Fan et al., 2024b) simplifies NPO by removing the reference model. FLAT (Wang et al., 2024) reframes unlearning as maximizing the f-divergence between forget and retain sets. RULE

(Zhang et al., 2025) integrates reinforcement learning by combining SFT with a subsequent RL phase. In summary, GA focuses solely on maximizing the next-token loss over the forget set, which can lead to severe utility collapse. GD alleviates this issue by incorporating retain data, but such data are often difficult to obtain and may be semantically unrelated to the forget set. To address these limitations, we propose **SGA**, which leverages a normal model to generate a customized normal set, thereby achieving better forgetting quality and model utility during unlearning.

## 3 PRELIMINARIES

### 3.1 FORMULATION

Given a pre-trained dataset $\mathcal{D}$, we obtain a pre-trained large language model $\pi_{\text{origin}}$. The goal of LLM unlearning is to enable $\pi_{\text{origin}}$ to forget hazardous knowledge while preserving its normal performance as much as possible. In fine-tuning-based unlearning methods, we update the model parameters $\theta$ through training, thereby transforming $\pi_{\text{origin}}$ into $\pi_{\text{unlearn}}$.

In unlearning tasks, it is common to construct a forget dataset $\mathcal{D}_f$ and a retain dataset $\mathcal{D}_r$ from the pre-training corpus $\mathcal{D}$. Here, $\mathcal{D}_f$ denotes the data that $\pi_{\text{origin}}$ should forget, which is used to evaluate the forgetting performance of unlearning methods; while $\mathcal{D}_r$ denotes the data that $\pi_{\text{origin}}$ should retain, which is used to assess the preservation of the model's original capabilities. The parameter $\lambda$ serves as a regularization coefficient to balance these two objectives. Let $x$ denote the model input and $y$ the next-token output, where subscripts $f$ and $r$ indicate samples drawn from the forget set and the retain set, respectively. Formally, the optimization objective of unlearning methods can be expressed as:

$$\min_{\theta} \underbrace{\mathbb{E}_{(x_f, y_f) \in \mathcal{D}_f} \big[ \ell_f(y_f \mid x_f; \theta) \big]}_{\text{forget}} + \lambda \underbrace{\mathbb{E}_{(x_r, y_r) \in \mathcal{D}_r} \big[ \ell_r(y_r \mid x_r; \theta) \big]}_{\text{retain}}. \tag{1}$$

### 3.2 GRADIENT ASCENT

The conventional Gradient Ascent (GA) method performs unlearning solely on the forget data, which corresponds to setting $\lambda = 0$ in Equation 1:

$$\min_{\theta} \underbrace{\mathbb{E}_{(x_f, y_f) \in \mathcal{D}_f} \big[ \ell_f(y_f \mid x_f; \theta) \big]}_{\text{forget}}. \tag{2}$$

GA achieves unlearning by negating the original training loss, thereby reversing the learning process into "forgetting" and diminishing the influence of previously learned data. At step $t$, GA updates the model in the direction that *maximizes* the next-token prediction loss on the forget set (Wang et al., 2024; Yao et al., 2024), i.e., $\theta_{t+1} \leftarrow \theta_t + \lambda \nabla_{\theta_t} L(x, y; \theta_t)$, where $\lambda$ denotes the (un)learning rate. However, GA is prone to catastrophic collapse (Zhang et al., 2024a) due to its inherently divergent nature.

To address this, we hypothesize that instead of relying solely on forget data, the model can also be provided with safe answers. While GA maximizes the loss on the forget set, the additional safe losses can partially redirect its optimization trajectory. Based on this intuition, we propose **SGA**, a gradient combination approach inspired by generalized label smoothing.

### 3.3 GENERALIZED LABEL SMOOTHING

Generalized Label Smoothing (GLS) is a simple yet effective learning paradigm that has been widely applied in trustworthy machine learning and deep learning (Wei et al., 2021; Szegedy et al., 2016; Lukasik et al., 2020; Liu & Guo, 2020). Let $y_i$ be the one-hot encoded vector form of the label generated according to $Y$. The random variable of the generalized smoothed label $Y^{\text{GLS},r}$ with smooth rate $r \in (-\infty, 1]$ is defined as $y_i^{\text{GLS},r} := (1-r) \cdot y_i + \frac{r}{K} \cdot \mathbf{1}$ where $K$ is the number of classes and $\mathbf{1}$ is the all-ones vector. For example, when $r = 0.3$ and $y_i = [1, 0, 0]^\top$, the generalized smoothed label becomes $y_i^{\text{GLS},0.3} = [0.8, 0.1, 0.1]^\top$. Conversely, when $r = -0.3$, it becomes

$y_i^{\text{GLS},-0.3} = [1.2, -0.1, -0.1]^\top$. It is worth noting that previous studies have suggested that Label Smoothing can be applied to Machine Unlearning(Di et al., 2024b). However,our work is the first to apply Label Smoothing in the context of LLM Unlearning.

# 4 SMOOTHED GRADIENT ASCENT (SGA)

In this section, we provide a detailed description of the **SGA** method, followed by an analysis of its improvements over the standard GA approach. Finally, we examine the potential values for estimating the optimal smooth rate.

## 4.1 METHOD DETAILS

**Step1: Normal Data Generation.** For each forget data instance, we generate $K-1$ corresponding normal data samples. The generation of normal data follows the principle that they should either be semantically similar to the forget data or constitute entirely safe responses. For example, in the TOFU experiments, we select the $K-1$ samples from the retain data that have the highest cosine similarity in embeddings with the forget data and exceed a predefined threshold; if such samples cannot be found, we substitute them with safe responses such as "I don't know." In the MUSE and Harry Potter experiments, we employ GPT-4o-mini (Achiam et al., 2023) to generate $K-1$ semantically similar normal data samples for each forget data instance, ensuring that they do not contain any harmful information. We present the procedure for generating normal data with GPT-4o-mini in Appendix D.

**Step 2: Smoothed Gradient Ascent.** We extend the idea of **Generalized Label Smoothing (GLS)** by introducing a smoothing rate $r$, which combines the forget data with $K-1$ normal data samples to guide the base LLM during the unlearning and learning process. Under the standard GA setting (**SGA** with $r=0$), the model ignores the normal data. In the $K$-dimensional label space (forget data, normal data$_1$, normal data$_2$, ...), the target label is $(-1, 0, 0, 0, ...)$, where the negative coefficient indicates that the model is driven to **forget** the corresponding data, while positive coefficients represent reinforcement through **learning**. In contrast, under **SGA** with a smoothing rate $r$, the label distribution becomes $\left(-\left(1 - \frac{K-1}{K}r\right), -\frac{r}{K}, -\frac{r}{K}, ...\right)$. Finally, the optimization objective in (1) can be reformulated as:

$$\min_\theta \underbrace{\left(1 - r + \tfrac{r}{K}\right) \mathbb{E}_{(x_f, y_f) \in \mathcal{D}_f}\left[\ell_f(y_f \mid x_f; \theta)\right]}_{\text{forget}} + \underbrace{\left(\tfrac{r}{K}\right) \mathbb{E}_{(x_p, y_p) \in \mathcal{D}_p}\left[\sum_{k=1}^{K} \ell_p^{(k)}(y_p^{(k)} \mid x_p^{(k)}; \theta)\right]}_{\text{normal data}}.$$
(3)

Here, the subscript $p$ denotes samples drawn from our generated normal set.

## 4.2 WHY SGA ALLEVIATES THE DIVERGENCE PROBLEM

From a gradient viewpoint, GA follows the ascent direction of the forget loss only:

$$\Delta\theta_t \propto -g_f, \qquad g_f \triangleq \nabla_{\theta_t} L_f.$$

This purely divergent direction often leads to catastrophic collapse.

Under our GLS-based construction (Equation 3), the per-step objective combines the forget loss with $K$ normal losses via the smooth rate $r$:

$$\mathcal{L}_{\text{SGA}}(\theta) = \left(1 - r + \tfrac{r}{K}\right) L_f(\theta) + \left(\tfrac{r}{K}\right) \sum_{k=1}^{K} L_p^{(k)}(\theta).$$
(4)

Taking the gradient yields the combined update direction

$$\Delta\theta_t \propto -\nabla_{\theta_t}\mathcal{L}_{\text{SGA}}(\theta_t) = -\left[\left(1 - r + \tfrac{r}{K}\right) g_f + \left(\tfrac{r}{K}\right) \sum_{k=1}^{K} g_p^{(k)}\right],$$
(5)

where $g_p^{(k)} \triangleq \nabla_{\theta_t} L_p^{(k)}$ denotes the gradient contributed by the $k$-th normal sample. Thus, a key reason why **SGA** effectively suppresses the divergence issue of GA is that it alters GA's gradient-ascent direction, preventing the model from updating purely toward maximizing the next-token loss on the forget set.

### 4.3 OPTIMAL SMOOTH RATE

Given the forget data and $K - 1$ normal data, we hypothesize the existence of an optimal smoothing rate $r^*$. Let $g_f$ denote the gradient of the forget loss and $g_p^{(k)}$ the gradient of the $k$-th normal loss, with their average denoted as $\bar{g}_p$.

The one-step update vector is:

$$d(r) = -g_f + r\left[\bar{g}_p - \left(1 - \tfrac{1}{K}\right)g_f\right], \tag{6}$$

where $r$ provides a tunable deflection from the GA direction $-g_f$ along

$$u \triangleq \bar{g}_p - \left(1 - \tfrac{1}{K}\right)g_f. \tag{7}$$

Without this adjustment, GA updates purely along $-g_f$, often causing divergence. To mitigate this, we seek to minimize the update norm while ensuring forgetting:

$$r^* = \arg\min_r \|d(r)\|^2. \tag{8}$$

Solving yields the closed-form:

$$r^* = \frac{\langle g_f, u \rangle}{\|u\|^2}. \tag{9}$$

**Discussion on Equation 9.** Equation 9 indicates that the optimal $r^*$ is jointly determined by the base LLM, the forget data, and the normal data. Since the model parameters $\theta$ evolve during unlearning, $r^*$ also changes dynamically. For efficiency, we fix $r$ at the start of training, so the empirical optimum in Section 5 cannot be derived directly from Equation 9. Nevertheless, by estimating the sign of $\langle g_f, u \rangle$ on the base LLM, which characterizes the feasible range of $r^*$, we find:

- $\langle g_f, u \rangle > 0$ (the angle between $g_f$ and $u$ is less than $90°$): $r^*$ tends to be positive;
- $\langle g_f, u \rangle < 0$ (the angle between $g_f$ and $u$ is greater than $90°$): $r^*$ tends to be negative.

The effective ranges reported in Section 5.2 align with this estimation, as shown in Figure 2.

## 5 EXPERIMENT

In this section, we evaluate the performance of the **SGA** method on three established LLM unlearning tasks: TOFU (Maini et al., 2024), MUSE-News (Shi et al., 2024), and Harry Potter (Yao et al., 2024). Among them, the TOFU task focuses on the problem of entity unlearning, the Harry Potter experiment addresses copyright-related issues, and MUSE-News concerns the evaluation of general unlearning capabilities.

### 5.1 BASELINE METHODS

We compare **SGA** against other fine-tuning-based LLM unlearning methods across three tasks to evaluate its effectiveness. For all tasks, we include Gradient Ascent (GA) (Yao et al., 2024), KL minimization (KL) (Maini et al., 2024), GradDiff (GD) (Liu et al., 2022), NPO (Zhang et al., 2024a), and Forget-data-only Loss Adjustment (FLAT) (Wang et al., 2024) as baselines. On the copyrighted content task (Harry Potter) and the entity unlearning task (TOFU), we further compare with Preference Optimization (PO) (Maini et al., 2024), Large Language Model Unlearning (LLMU) (Yao et al., 2024), and DPO. For the Harry Potter task, we additionally include the Missmatch (Liu et al., 2024) method. For the MUSE-News task, we incorporate Task Vector, Who's Harry Potter (WHP) (Eldan & Russinovich, 2023), and NPO-RT (Zhang et al., 2024a) as baselines.

## 5.2 ENTITY UNLEARNING

**Experiment Setup.** The TOFU(Maini et al., 2024) dataset is a question-answering benchmark built on the biographies of 200 fictional authors, with each author associated with 20 QA pairs. Based on the designated unlearning scope, the dataset is partitioned into a forget set and a retain set. In our experiments, we adopt the 1% forget split. Following (Deng et al., 2025; Wang et al., 2024), we use Llama2-7B (Touvron et al., 2023), Phi-1.5B (Li et al., 2023), and OPT-2.7B (Zhang et al., 2022) as the base LLMs. In addition, we employ the bge-large-en-v1.5 embedding model (Xiao et al., 2023) as an auxiliary encoder to select normal data from the retain set for training.

**Evaluation Metrics.** To evaluate both the forgetting quality and the retained utility of the unlearned models, we employed two metrics from the TOFU benchmark: **Forget Quality (FQ)** and **Model Utility (MU)** (Maini et al., 2024). FQ is evaluated using the p-value of a Kolmogorov–Smirnov test comparing the unlearned model's outputs with those of a retain-only model, where higher values indicate stronger forgetting. MU reflects the model's utility by aggregating performance on held-out retain data spanning fictional authors, real-world profiles, and factual knowledge. We additionally report **ROUGE-L** scores on both the forget and retain sets. On the forget set, a score closer to that of the retain model indicates better forgetting performance, while on the retain set, higher ROUGE-L scores correspond to better utility. Full metric details are provided in Appendix C.1

Table 1: We evaluate the performance of **SGA** and baseline approaches on three base LLMs: Llama2-7B, OPT-2.7B, and Phi-1.5B. Here, FQ, MU, F-RL, and R-RL denote Forget Quality, Model Utility, and the ROUGE-L scores on the forget and retain sets, respectively. For reference, we also report results from the Original and Retained LLMs. The top-2 results are highlighted in blue .

| Base LLM | Llama2-7B | | | | OPT-2.7B | | | | Phi-1.5B | | | |
|---|---|---|---|---|---|---|---|---|---|---|---|---|
| Metric | FQ(↑) | MU(↑) | F-RL(↓) | R-RL(↑) | FQ(↑) | MU(↑) | F-RL(↓) | R-RL(↑) | FQ(↑) | MU(↑) | F-RL(↓) | R-RL(↑) |
| Original LLM | 4.4883e-06 | 0.6239 | 0.9851 | 0.9818 | 0.0013 | 0.5112 | 0.7537 | 0.8807 | 0.0013 | 0.5195 | 0.9607 | 0.9276 |
| Retained LLM | 1.0 | 0.6267 | 0.4080 | 0.9833 | 1.0 | 0.5067 | 0.4217 | 0.7669 | 1.0 | 0.5233 | 0.4272 | 0.9269 |
| GA/SGA (r = 0) | 0.0068 | 0.5990 | 0.4817 | 0.9204 | 0.0541 | 0.4851 | 0.4490 | 0.6440 | 0.0541 | 0.5058 | 0.4914 | 0.8012 |
| KL | 0.0030 | 0.5994 | 0.4922 | 0.9172 | 0.0143 | 0.4785 | 0.3971 | 0.6010 | 0.0541 | 0.5087 | 0.4910 | 0.7603 |
| GD | 0.0068 | 0.5998 | 0.4869 | 0.9182 | 0.0971 | 0.4826 | 0.4085 | 0.6016 | 0.0286 | 0.5117 | 0.4991 | 0.7959 |
| LLMU | 0.0030 | 0.5999 | 0.4891 | 0.9236 | 0.0068 | 0.1849 | 0.0210 | 0.1727 | 0.0143 | 0.5108 | 0.3331 | 0.7785 |
| PO | 0.0030 | 0.6323 | 0.1752 | 0.9169 | 0.0971 | 0.3706 | 0.0393 | 0.3594 | 0.0541 | 0.5064 | 0.4958 | 0.8003 |
| DPO-RT | 0.0068 | 0.6322 | 0.2595 | 0.9091 | 0.0068 | 0.3278 | 0.0348 | 0.2741 | 0.0286 | 0.5053 | 0.2824 | 0.7372 |
| NPO-RT | 0.0030 | 0.5994 | 0.5049 | 0.9270 | 0.0286 | 0.4844 | 0.4353 | 0.6401 | 0.0286 | 0.5090 | 0.4957 | 0.7660 |
| FLAT (Pearson) | 0.0030 | 0.6304 | 0.4825 | 0.9284 | 0.0030 | 0.4714 | 0.1962 | 0.4973 | 0.0541 | 0.5160 | 0.4228 | 0.7674 |
| **SGA** (r = 0.8) | 0.0030 | 0.6051 | 0.5352 | 0.8929 | 0.0971 | 0.4584 | 0.4360 | 0.5449 | 0.0143 | 0.5085 | 0.4963 | 0.8097 |
| **SGA** (r = 0.4) | 0.0068 | 0.6027 | 0.4792 | 0.9163 | 0.0286 | 0.4740 | 0.4033 | 0.5862 | 0.0143 | 0.5085 | 0.4963 | 0.8097 |
| **SGA** (r = 0.2) | 0.0068 | 0.6018 | 0.4777 | 0.9229 | 0.0286 | 0.4799 | 0.3967 | 0.5996 | 0.0286 | 0.5060 | 0.4782 | 0.8075 |
| **SGA** (r = -0.2) | 0.0068 | 0.6025 | 0.4909 | 0.9270 | 0.0971 | 0.4851 | 0.3973 | 0.6148 | 0.0971 | 0.5100 | 0.4831 | 0.8062 |
| **SGA** (r = -0.4) | 0.0068 | 0.6032 | 0.4888 | 0.9291 | 0.0971 | 0.4859 | 0.4021 | 0.6939 | 0.0971 | 0.5093 | 0.4851 | 0.8009 |
| **SGA** (r = -0.8) | 0.0030 | 0.6037 | 0.4871 | 0.9281 | 0.2657 | 0.4867 | 0.3946 | 0.6139 | 0.0971 | 0.5092 | 0.4957 | 0.8040 |
| **SGA** (r = -2) | 0.0030 | 0.6046 | 0.4897 | 0.9224 | 0.2657 | 0.4784 | 0.3959 | 0.6012 | 0.0971 | 0.5120 | 0.4816 | 0.8043 |
| **SGA** (r = -4) | 0.0030 | 0.6047 | 0.4985 | 0.9217 | 0.4046 | 0.4747 | 0.3863 | 0.5890 | 0.0971 | 0.5095 | 0.4891 | 0.7953 |
| **SGA** (r = -8) | 0.0013 | 0.6060 | 0.4916 | 0.9280 | 0.4046 | 0.4749 | 0.3895 | 0.5873 | 0.0541 | 0.5074 | 0.4877 | 0.7918 |

**SGA achieves the best forget quality across all models.** According to Table 1, we observe that, compared with all baseline methods, the **SGA** under the optimal smoothing rate achieves the best forget quality. Moreover, the forget quality of **SGA** substantially surpasses that of GA (i.e., **SGA** with a smoothing rate of 0), indicating that incorporating new normal data for continued learning while forgetting the target data leads to more effective unlearning.

**Compared with GA (corresponding to SGA with r=0), SGA generally demonstrates superior performance.** Across all three models, **SGA** with an appropriately tuned smooth rate consistently outperforms Gradient Ascent on nearly all evaluation metrics. This suggests that jointly incorporating normal data with forget data during training enables

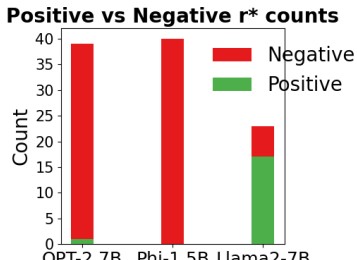

Figure 2: Following Section 4.3, we estimate the sign of $r^\star$ for each forget data instance in the TOFU benchmark forget01 dataset, which corresponds to the sign of $\langle g_f, u \rangle$ in Equation 9.

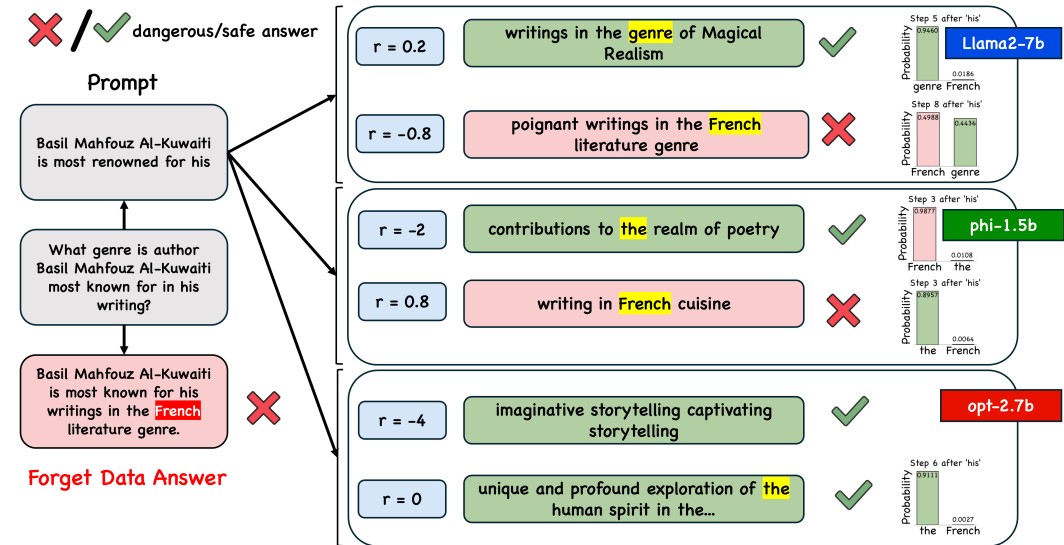

Figure 3: Different smoothing rates yield distinct output probabilities for critical tokens across base LLMs. As shown in the figure, the forget data highlights Basil Mahfouz Al-Kuwaiti's achievements in **French** literary style. In the LLaMA2-7B model, training with $r = -0.8$ assigns a much higher probability to the token "French," whereas training with $r = 0.2$ makes the model far less likely to produce this privacy-sensitive token.

the model to achieve effective forgetting while simultaneously preserving its performance to a greater extent.

**The validity of the optimal smooth rate $r^*$ range is verified on the TOFU benchmark.** According to Section 4.3, we estimate the sign of $\langle g_f, u \rangle$ for each data instance and each base LLM in the TOFU experiments, as illustrated in Figure 2. Compared with Table 1, we observe that the empirically optimal smooth rate values largely align with the estimation results. Importantly, different smooth rates can substantially alter the model's confidence in generating certain key pieces of information, thereby enabling unlearning, as illustrated in Figure 3.

### 5.3 COPYRIGHTED CONTENT UNLEARNING

**Experiment Setup.** Following prior work (Deng et al., 2025; Wang et al., 2024), we use *Harry Potter and the Sorcerer's Stone* (Rowling, 2023; Eldan & Russinovich, 2023) as copyrighted material for unlearning. We extract 400 segments ($\leq 512$ tokens each) as the forget set $\mathcal{D}_f$ (Deng et al., 2025; Wang et al., 2024; Jia et al., 2024b), and sample 400 paragraphs from C4 (Raffel et al., 2020) as the retain set $\mathcal{D}_r$. For each forget instance, GPT-4o-mini (Achiam et al., 2023) generates safe normal data. Following (Wang et al., 2024), we fine-tune OPT-2.7B (Zhang et al., 2022) and Llama2-7B (Touvron et al., 2023) to simulate memorization and run experiments on the fine-tuned models.

**Evaluation Metrics.** Following the choice of evaluation metrics in prior work (Deng et al., 2025; Wang et al., 2024), we also consider both unlearning effectiveness and model utility. Unlearning effectiveness is measured using the Forget Quality Gap (FQ Gap), which evaluates the forgetting performance based on the differences in BLEU (Papineni et al., 2002) and ROUGE-L (Lin, 2004) scores between the unlearned model and the retained model. Model utility is assessed through perplexity (PPL) on Wikitext (Merity et al., 2016) and the average accuracy across nine zero-shot benchmarks. Full metric definitions are provided in Appendix C.2.

**By appropriately tuning the smoothing rate, SGA consistently outperforms Gradient Ascent.** As shown in Table 2, aside from a few cases where training collapses under extreme smoothing rates, **SGA** consistently surpasses GA (i.e., **SGA** with $r = 0$) across all evaluation metrics, including FQ Gap, PPL, and Avg. ACC. This corroborates our discussion in Section 4.2, where we argued that incorporating normal data effectively mitigates GA's instability, which otherwise maximizes loss

Table 2: **SGA** and baseline methods are evaluated on the Harry Potter benchmark using two base LLMs: OPT-2.7B and Llama2-7B. We report only the best-performing $r$ for each base model, with full results provided in Appendix F.1. Among the three evaluation metrics—Forget Quality Gap (FQ Gap), Perplexity (PPL), and Average Accuracy (Avg. Acc.)—the top-2 scores are highlighted in blue. Values marked with * are taken directly from (Wang et al., 2024).

| Base LLM | OPT-2.7B | | | Llama2-7B | | |
|---|---|---|---|---|---|---|
| Metric | FQ Gap($\downarrow$) | PPL($\downarrow$) | Avg. Acc.($\uparrow$) | FQ Gap($\downarrow$) | PPL($\downarrow$) | Avg. Acc.($\uparrow$) |
| Original LLM | 1.5346 | 15.6314 | 0.4762 | 3.6594 | 8.9524 | 0.5617 |
| Retained LLM | 0.0 | 14.3190 | 0.4686 | 0.0 | 8.7070 | 0.5599 |
| GA/SGA (r=0)* | 2.7301 | 1.0984e71 | 0.3667 | 0.4587 | 47.2769 | 0.5088 |
| KL* | 2.7301 | 16.1592 | 0.4688 | 0.4225 | 9.4336 | 0.5509 |
| GD* | 2.3439 | 16.1972 | 0.4690 | 0.5304 | 9.1797 | 0.4902 |
| Mismatch* | 1.4042 | 15.7507 | 0.4679 | 0.4647 | 8.9906 | 0.5593 |
| LLMU* | 2.4639 | 15.8398 | 0.4656 | 0.1985 | 9.0530 | 0.5503 |
| PO* | 2.1601 | 14.8960 | 0.4583 | 0.5124 | 8.8364 | 0.5532 |
| DPO* | 2.2152 | 16.8396 | 0.4621 | 0.2924 | 8.9597 | 0.5614 |
| NPO* | 1.2611 | 19.6637 | 0.4644 | 0.5151 | 9.0397 | 0.5609 |
| FLAT (Pearson)* | 1.4089 | 15.5543 | 0.4686 | 0.2265 | 8.9906 | 0.5580 |
| **SGA** (r=-0.2) | 0 | 6.2700e72 | 0.3683 | 0.0761 | 28.9249 | 0.5421 |
| **SGA** (r=-0.4) | 0 | 6.8530e30 | 0.3876 | 0.5355 | 11.4885 | 0.5676 |
| **SGA** (r=-0.8) | 0.0043 | 1.3040e6 | 0.4497 | 0.4141 | 19.9494 | 0.5316 |
| **SGA** (r=-2) | 0.0084 | 6.7100e5 | 0.4532 | 0.350 | 18.67 | 0.5305 |

solely on the forget set. Notably, on OPT-2.7B and Llama2-7B, the GA baseline suffers severe divergence, manifested as extremely high PPL and drastically low Avg. ACC. In contrast, **SGA** substantially suppresses this issue, rendering GA's divergence problem far less severe.

**SGA generally achieves superior forgetting capability.** On OPT-2.7B and Llama2-7B, we observe that when the smoothing rate is properly tuned, **SGA** typically attains the strongest forgetting performance. This advantage arises because **SGA** still applies gradient ascent on the forget data as part of its unlearning process, thereby inheriting GA's beneficial aspects while avoiding instability.

**Discussion on Perplexity (PPL).** For the OPT-2.7B and Llama2-7B models, GA yields extremely high PPL values. This indicates that models after GA-based unlearning exhibit very high perplexity, reflecting the severe divergence of GA and the resulting degradation in model utility. In contrast, the PPL results for **SGA** in Table 2 demonstrate that it significantly alleviates this issue, even reducing the PPL on Llama2-7B to a level comparable with other unlearning methods. This further confirms that by jointly leveraging normal data and forget data, **SGA** effectively mitigates the divergence problem inherent to GA.

## 5.4 MUSE-NEWS UNLEARNING

**Experiment Setup.** We evaluate **SGA** on the MUSE-News benchmark (Shi et al., 2024), constructed from real-world BBC news articles and partitioned into three subsets: a forget set, a retain set, and a holdout set for utility evaluation. In experiments, we perform unlearning on the Llama2-7B (Touvron et al., 2023) pretrained model provided by the MUSE benchmark. The baseline methods in the evaluation are obtained or reproduced from the original implementations provided by MUSE, while **SGA** is specifically developed and implemented on the MUSE-News benchmark.

**Evaluation Metrics.** We evaluate both the baseline methods and **SGA** using the metrics provided by the MUSE benchmark: *VerbMem* on forget dataset, *KnowMem* on forget and retain dataset, and Privacy leakage (*PrivLeak*). *VerbMem* measures the model's ability to continue generating forgotten text, while *KnowMem* evaluates whether the model preserves knowledge from both the forget and retain sets. *PrivLeak* assesses privacy leakage via membership inference (MIA). For a more detailed description of the evaluation metrics used in MUSE-NEWS, please refer to the Appendix C.3

**SGA maintains GA's strong performance on the forget set** $D_f$ **while improving upon it on the retain set** $D_r$**.** On $D_f$, **SGA** substantially reduces memorization risk in VerbMem and KnowMem, achieving scores of 0 across both metrics—well below the Retained LLM baseline and fully com-

Table 3: The evaluation on the MUSE benchmark is conducted across four criteria. Results are highlighted in  blue  when the unlearning method satisfies the criterion, and in  red  when it does not. For $D_f$, smaller values are preferable, whereas for $D_r$, larger values are desirable. For PrivLeak, the ideal outcome is close to 0, since substantial deviations in either direction may indicate privacy leakage. We further highlight in  green  the top-2 PrivLeak results that also meet the other three criteria. Only the best-performing $r$ for each method is shown here, with full results provided in Appendix F.2. Values marked with * are taken directly from (Wang et al., 2024).

| | VerbMem on $D_f$ ($\downarrow$) | | KnowMem on $D_f$ ($\downarrow$) | | KnowMem on $D_r$ ($\uparrow$) | | PrivLeak |
|---|---|---|---|---|---|---|---|
| Original LLM | 58.4 | - | 63.9 | - | 55.2 | - | -99.8 |
| Retained LLM | 20.8 | - | 33.1 | - | 55.0 | - | 0.0 |
| Task Vectors* | 56.3 | (✘) | 63.7 | (✘) | 54.6 | (✔) | -99.8 |
| WHP* | 19.7 | (✔) | 21.2 | (✔) | 28.3 | (✔) | 109.6 |
| GA* | 0.0 | (✔) | 0.0 | (✔) | 0.0 | (✘) | 17.0 |
| GD* | 4.9 | (✔) | 27.5 | (✔) | 6.7 | (✔) | 109.4 |
| KL* | 27.4 | (✘) | 50.2 | (✘) | 44.8 | (✔) | -96.1 |
| NPO* | 0.0 | (✔) | 0.0 | (✔) | 0.0 | (✘) | 15.0 |
| NPO-RT* | 1.2 | (✔) | 54.6 | (✘) | 40.5 | (✔) | 105.8 |
| FLAT (Pearson)* | 1.6 | (✔) | 0.0 | (✔) | 0.2 | (✔) | 26.8 |
| **SGA** | 0 | (✔) | 0 | (✔) | 1.9498 | (✔) | **15.5700** |

parable to GA. On $D_r$, **SGA** raises GA's KnowMem score from 0 to 1.9498, thereby meeting the criterion that KnowMem should not be 0. Consequently, **SGA** not only inherits GA's strengths on $D_f$ but also effectively remedies its shortcomings on $D_r$.

**Among methods satisfying the criteria on both VerbMem and KnowMem, SGA ranks within the top-2 for PrivLeak, achieving a score of 15.57.** This demonstrates that **SGA** provides substantially better control over privacy leakage compared to other approaches. In summary, on the MUSE-NEWS benchmark, **SGA** has been validated as an unlearning method that not only meets all evaluation criteria but also achieves the strongest performance in mitigating privacy leakage.

## 5.5 ABLATION STUDIES

**Ablation study on the selection of TOFU normal data**  As described in Section 5.2, the procedure for collecting normal data in the TOFU experiment differs from the other two experiments. Instead of relying entirely on generation from an external large model, we compute the similarity between the retain data and forget data using an embedding model, and then select the most similar samples from the retain set as normal data. In the ablation study, we compare the results of training with normal data generated by GPT-4o-mini against those obtained using normal data selected by the embedding model.

**The SGA method supported by GPT does not lose its effectiveness;**  instead, it even achieves better performance than normal data generated by the embedding model on certain base LLMs. However, we also observe that as the source of normal data changes, the value of the optimal smoothing rate shifts accordingly, which is consistent with our analysis in Section 4.3.

**Ablation study on the selection of external LLMs**  To avoid potential bias introduced by using the GPT-4o-mini model for generating normal data, we replaced it with Qwen3-Max(Yang et al., 2025) and DeepSeek-V3.2(DeepSeek-AI et al., 2025), and re-conducted the experiments on the TOFU benchmark. The results are presented in table 5. From the results, we can observe that changing the external LLM does not affect the effectiveness of the SGA method.

**Ablation study on the value of $K$**  To study the impact of the parameter $K$ on the unlearning performance, we conducted an ablation experiment. The results indicate that the value of $K$ does not substantially affect the unlearning outcome. In practical applications, we recommend selecting $K$ according to specific needs. It is worth noting that $K$ should be greater than 1; when $K = 1$, the

Table 4: Ablation study results on TOFU across different base LLMs. We compare **SGA** selecting normal data via an embedding model (**SGA**) with **SGA** employing GPT-4o-mini to generate normal data (GPT), in terms of Forget Quality (FQ) and Model Utility (MU). For each method, we report results for the two best smoothing rates, with the top-2 outcomes highlighted in  blue .

| Llama2-7B | | | OPT-2.7B | | | Phi-1.5B | | |
|---|---|---|---|---|---|---|---|---|
| Metric | FQ(↑) | MU(↑) | Metric | FQ(↑) | MU(↑) | Metric | FQ(↑) | MU(↑) |
| **SGA** (r=0.4) | 0.0068 | 0.6027 | **SGA** (r=-0.8) | 0.2657 | 0.4867 | **SGA** (r=-0.8) | 0.0971 | 0.5092 |
| **SGA** (r=0.2) | 0.0068 | 0.6018 | **SGA** (r=-2) | 0.2657 | 0.4784 | **SGA** (r=-2) | 0.0971 | 0.5120 |
| GPT (r=-2) | 0.1650 | 0.6170 | GPT (r=0.4) | 0.2657 | 0.4830 | GPT (r=-0.4) | 0.0068 | 0.4594 |
| GPT (r=-4) | 0.2657 | 0.6189 | GPT (r=-8) | 0.2657 | 0.4822 | GPT (r=-0.2) | 0.0068 | 0.4669 |

Table 5: Ablation study results on the selection of external LLMs for generating normal data on the TOFU benchmark. We compare results using GPT-4o-mini, Qwen3-Max, and DeepSeek-V3.2 across different base LLMs (Llama2-7B, OPT-2.7B, Phi-1.5B), in terms of Forget Quality (FQ) and Model Utility (MU).

| Llama2-7B | | | OPT-2.7B | | | Phi-1.5B | | |
|---|---|---|---|---|---|---|---|---|
| External LLM | FQ(↑) | MU(↑) | External LLM | FQ(↑) | MU(↑) | External LLM | FQ(↑) | MU(↑) |
| GPT-4o-mini (r = -2) | 0.1650 | 0.6170 | GPT-4o-mini (r = 0.4) | 0.2657 | 0.4830 | GPT-4o-mini (r = -0.4) | 0.0068 | 0.4594 |
| GPT-4o-mini (r = -4) | 0.2657 | 0.6189 | GPT-4o-mini (r = -8) | 0.2657 | 0.4822 | GPT-4o-mini (r = -0.2) | 0.0068 | 0.4669 |
| Qwen3-Max (r = -2) | 0.1650 | 0.6216 | Qwen3-Max (r = -0.2) | 0.1650 | 0.4735 | Qwen3-Max (r = -0.2) | 0.1650 | 0.6331 |
| DeepSeek-V3.2 (r = -2) | 0.1650 | 0.6331 | DeepSeek-V3.2 (r = -0.2) | 0.0971 | 0.4730 | DeepSeek-V3.2 (r = -0.2) | 0.1650 | 0.4938 |

number of normal data samples becomes $K - 1 = 0$, and our method degenerates into **Gradient Ascent**. The results of the $K$ ablation study are presented in 6.

Table 6: Ablation study on the choice of $K$ for the proposed method. We evaluate different values of $K$ on the TOFU benchmark across three base LLMs (Llama2-7B, OPT-2.7B, and Phi-1.5B), reporting Forget Quality (FQ) and Model Utility (MU).

| | Llama2-7B | | OPT-2.7B | | Phi-1.5B | |
|---|---|---|---|---|---|---|
| External LLM | FQ(↑) | MU(↑) | FQ(↑) | MU(↑) | FQ(↑) | MU(↑) |
| SGA (K = 2) | 0.1650 | 0.6037 | 0.0971 | 0.4732 | 0.1650 | 0.4945 |
| SGA (K = 4) | 0.0030 | 0.6037 | 0.2657 | 0.4867 | 0.0971 | 0.5092 |
| SGA (K = 6) | 0.1650 | 0.6153 | 0.0286 | 0.4699 | 0.2657 | 0.4945 |

## 6 CONCLUSION

This paper addresses the limitations of Gradient Ascent in LLM unlearning and introduces **Smoothed Gradient Ascent (SGA)**. Inspired by Generalized Label Smoothin, **SGA** leverages auxiliary normal models to generate customized normal data for the forget set and integrates them via a tunable smoothing rate $r$, enabling the base LLM to jointly learn and unlearn, thereby effectively "forgetting" hazardous information. We provide a theoretical analysis of the optimal smoothing rate $r^*$ and empirically validate its feasible range. Experiments on TOFU, Harry Potter, and MUSE-NEWS show that **SGA** consistently outperforms Gradient Ascent and a list of strong baselines, delivering stronger Forget Quality, mitigating catastrophic divergence in Model Utility, and achieving state-of-the-art results on several key metrics.

## ETHICS STATEMENT AND REPRODUCIBILITY STATEMENT

**Ethics statement.** This work does not involve any ethical or moral concerns.

**Reproducibility statement.** The computational resources and experimental setup required for this study are provided in Appendix E. Upon acceptance of the paper, we will release all source code associated with our experiments.

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

APPENDIX

The Appendix is organized as follows.

- **Section A**: Presents the Broader Impacts and Limitations.
- **Section B**: Introduces the baseline methods compared in our experiments.
- **Section C**: Describes the evaluation metrics employed in the experiments.
- **Section D**: Demonstrates how we generate normal data using GPT-4o-mini.
- **Section E**: Details our experimental setup.
- **Section F**: Presenting our complete experimental results
- **Section G**: Outlines the extent to which we employ LLMs.

## A   BROADER IMPACTS AND LIMITATIONS

### A.1   BROADER IMPACTS

We propose an LLM unlearning method: **SGA**, that facilitates the removal of private, copyrighted, and offensive content that large models may have been exposed to during pre-training, thereby enhancing the safety and trustworthiness of LLMs. Moreover, we introduce a novel solution to mitigate the divergence issue inherent in the Gradient Ascent approach: incorporating the learning of customized safe information alongside the forgetting process. We observe that this not only strengthens the model utility after unlearning but also leads to significantly improved forgetting quality.

### A.2   LIMITATIONS

This paper argues that the optimal smoothing rate $r^*$ is highly dependent on the base LLM, the forget data, and the normal data, and that this theoretical optimum continuously evolves as model training progresses. Consequently, open questions remain as, whether dynamically adjusting the smoothing rate during the unlearning process could yield better results. These directions warrant further exploration in future work.

## B   BASELINE METHODS

**Gradient Ascent (GA) (Yao et al., 2024).**   Gradient Ascent (GA) is a simple baseline commonly adopted in machine unlearning. The key idea is to reverse the effect of gradient descent by maximizing the prediction loss on the forget dataset $\mathcal{D}_f$. This encourages the model to move away from knowledge associated with $\mathcal{D}_f$, thereby approximating a model trained only on the retain set $\mathcal{D}_r$. The objective is defined as:

$$\mathcal{L}_{\text{GA}} = -\frac{1}{|\mathcal{D}_f|} \sum_{(x_i, y_i) \in \mathcal{D}_f} \ell(x_i, y_i; \theta). \tag{10}$$

**Gradient Difference (GD) (Liu et al., 2024).**   Gradient Difference (GD) is another simple baseline for machine unlearning . Unlike GA, which only maximizes the loss on the forget set, GD combines two opposing objectives: minimizing the loss on the retain set $\mathcal{D}_r$ while maximizing the loss on the forget set $\mathcal{D}_f$. This dual objective encourages the model to preserve useful knowledge from $\mathcal{D}_r$ while forgetting information specific to $\mathcal{D}_f$. The composite loss is given by:

$$\mathcal{L}_{\text{GD}} = \frac{1}{|\mathcal{D}_r|} \sum_{(x_r, y_r) \in \mathcal{D}_r} \ell(x_r, y_r; \theta) - \frac{1}{|\mathcal{D}_f|} \sum_{(x_f, y_f) \in \mathcal{D}_f} \ell(x_f, y_f; \theta). \tag{11}$$

**KL Minimization (KL) (Maini et al., 2024).**   KL combines gradient ascent on the forget set $\mathcal{D}_f$ with a KL-divergence regularization on the retain set $\mathcal{D}_r$. Specifically, the model is encouraged to forget by maximizing the loss on $\mathcal{D}_f$, while its outputs on $\mathcal{D}_r$ are constrained to stay close to the

original (pre-unlearning) model $M_{\hat{\theta}}$ through Kullback–Leibler divergence. The objective can be formulated as:

$$\mathcal{L}_{\text{KL}} = -\frac{1}{|\mathcal{D}_f|} \sum_{(x_f, y_f) \in \mathcal{D}_f} \ell(x_f, y_f; \theta) + \frac{1}{|\mathcal{D}_r|} \sum_{x_r \in \mathcal{D}_r} \text{KL}\big(h_{\hat{\theta}}(x_r) \,\|\, h_{\theta}(x_r)\big), \qquad (12)$$

where $h_{\hat{\theta}}$ and $h_{\theta}$ denote the output distributions of the original and unlearned models, respectively.

**Preference Optimization (PO) (Maini et al., 2024).** PO modifies the model's response preferences by training it to output safe refusals (e.g., "I don't know") for prompts in the forget set $\mathcal{D}_f$. To achieve this, an augmented forget dataset $\mathcal{D}_{\text{IDK}}$ is constructed, where each input from $\mathcal{D}_f$ is paired with a refusal response. The overall objective combines the fine-tuning loss on the retain set $\mathcal{D}_r$ with a custom loss on $\mathcal{D}_{\text{IDK}}$:

$$\mathcal{L}_{\text{PO}} = \frac{1}{|\mathcal{D}_r|} \sum_{(x_r, y_r) \in \mathcal{D}_r} \ell(x_r, y_r; \theta) + \frac{1}{|\mathcal{D}_{\text{IDK}}|} \sum_{(x_f, y_{\text{IDK}}) \in \mathcal{D}_{\text{IDK}}} \ell(x_f, y_{\text{IDK}}; \theta). \qquad (13)$$

This encourages the model to retain performance on $\mathcal{D}_r$ while learning to reject answering queries from $\mathcal{D}_f$.

**Direct Preference Optimization (DPO) (Rafailov et al., 2023).** Direct Preference Optimization (DPO) adapts the preference optimization framework to the unlearning setting. Instead of comparing human-preferred and less-preferred responses, DPO contrasts a safe refusal completion $y_e$ with the original (to-be-forgotten) response $y_f$ under the same forget prompt $x_f \in \mathcal{D}_f$. The objective encourages the model to prefer $y_e$ over $y_f$, thereby enforcing targeted forgetting while preserving overall utility. Formally, with inverse temperature $\beta$, the loss is:

$$\mathcal{L}_{\text{DPO}} = -2\beta \, \mathbb{E}_{x_f \in \mathcal{D}_f} \left[ \log \sigma\Big(\beta \log h_{\theta}(x_f, y_e) \,-\, \beta \log h_{\theta}(x_f, y_f) - M_{\text{ref}}\Big) \right], \qquad (14)$$

where $h_{\theta}$ denotes the model's predictive distribution and $M_{\text{ref}}$ is an optional regularization term penalizing deviation from the original model. A retention-regularized variant further adds supervised loss on $\mathcal{D}_r$ to maintain desirable knowledge:

$$\mathcal{L}_{\text{DPO-RT}} = \mathcal{L}_{\text{DPO}} + \mathcal{L}(\mathcal{D}_r; \theta). \qquad (15)$$

**Negative Preference Optimization (NPO) (Zhang et al., 2024a).** Negative Preference Optimization (NPO) focuses on suppressing undesired responses by penalizing the likelihood of completions from the forget set $\mathcal{D}_f$. Unlike DPO, which contrasts preferred and dispreferred responses, NPO uses only the dispreferred term, directly discouraging the model from producing the original (to-be-forgotten) outputs. Formally, with inverse temperature $\beta$, the loss is:

$$\mathcal{L}_{\text{NPO}} = -2\beta \, \mathbb{E}_{(x_f, y_f) \in \mathcal{D}_f} \left[ \log \sigma\big(-\beta \log h_{\theta}(y_f | x_f)\big) \right], \qquad (16)$$

where $h_{\theta}$ denotes the model's predictive distribution. To preserve utility, a retention-regularized variant further adds supervised training on $\mathcal{D}_r$:

$$\mathcal{L}_{\text{NPO-RT}} = \mathcal{L}_{\text{NPO}} + \mathcal{L}(\mathcal{D}_r; \theta). \qquad (17)$$

**Mismatch.** Mismatch extends the preference-optimization framework by introducing randomly constructed text sequences (Yao et al., 2024). Similar to PO, it minimizes the fine-tuning loss on the retain set $\mathcal{D}_r$, but additionally incorporates a mismatch loss computed over a random combination of responses $Y_{\text{rnd}}$. This encourages the model to produce neutral outputs when exposed to random or irrelevant continuations, thus reinforcing unlearning. The objective is:

$$\mathcal{L}_{\text{Mismatch}} = \frac{1}{|\mathcal{D}_r|} \sum_{(x_r, y_r) \in \mathcal{D}_r} \ell(x_r, y_r; \theta) + \frac{1}{|\mathcal{D}_{\text{rnd}}|} \sum_{y_{\text{rnd}} \in Y_{\text{rnd}}} \ell(x_f, y_{\text{rnd}}; \theta), \qquad (18)$$

where $Y_{\text{rnd}}$ denotes a set of randomly sampled responses paired with forget prompts $x_f \in \mathcal{D}_f$.

**LLMU (Yao et al., 2024).** LLMU extends Gradient Ascent by incorporating two auxiliary components: (1) random-completion unlearning using sequences generated from forget prompts, and (2) retention regularization on normal retain data. The objective encourages forgetting by maximizing loss on the forget set $\mathcal{D}_f$, while simultaneously training on random completions $\mathcal{D}_{\text{rand}}$ and aligning the model's predictions on the retain set $\mathcal{D}_{\text{normal}}$ with the original model through KL divergence. Formally, the loss is:

$$\mathcal{L}_{\text{LLMU}} = - \frac{\varepsilon_1}{|\mathcal{D}_f|} \sum_{(x_f, y_f) \in \mathcal{D}_f} \ell(x_f, y_f; \theta)$$
$$+ \frac{\varepsilon_2}{|\mathcal{D}_{\text{rand}}|} \sum_{x \in \mathcal{D}_{\text{rand}}} \ell(x; \theta) \qquad (19)$$
$$+ \frac{\varepsilon_3}{|\mathcal{D}_{\text{normal}}|} \sum_{x \in \mathcal{D}_{\text{normal}}} \text{KL}\big(h_{\hat{\theta}}(x) \,\|\, h_\theta(x)\big) ,$$

**Task Vectors (Eldan & Russinovich, 2023).** The Task Vector method constructs an unlearned model by explicitly subtracting the adaptation direction on the forget set $\mathcal{D}_f$. Let $\theta_o$ denote the parameters of the original model and $\theta_{\text{reinforce}}$ the parameters of the model fine-tuned to overfit $\mathcal{D}_f$. The task vector is defined as the difference $(\theta_{\text{reinforce}} - \theta_o)$, and the unlearned model parameters are obtained by reversing this direction:

$$\theta = \theta_o - (\theta_{\text{reinforce}} - \theta_o). \qquad (20)$$

This procedure moves the model away from the representation adapted to $\mathcal{D}_f$ without requiring further optimization.

where $\varepsilon_1, \varepsilon_2, \varepsilon_3$ are weighting coefficients, and $h_{\hat{\theta}}$ denotes the output distribution of the original model.

**Who's Harry Potter (WHP) (Eldan & Russinovich, 2023).** WHP defines the unlearned model through a distributional interpolation between the original model $\theta_o$ and the reinforced model $\theta_{\text{reinforce}}$. For any input $x$, let $p_\theta(\cdot|x)$ denote the token-level output distribution of the unlearned model. WHP adjusts this distribution by subtracting a scaled task-adaptation direction:

$$p_\theta(\cdot|x) = p_{\theta_o}(\cdot|x) - \alpha\big(p_{\theta_{\text{reinforce}}}(\cdot|x) - p_{\theta_o}(\cdot|x)\big), \qquad (21)$$

where $\alpha$ is a tunable coefficient controlling the degree of forgetting. This effectively pushes the model's output distribution away from $p_{\theta_{\text{reinforce}}}$ while retaining alignment with $p_{\theta_o}$.

**FLAT (Wang et al., 2024).** Forget data only Loss AdjustmenT (FLAT) is a loss adjustment-based unlearning method that eliminates the need for retain data or a reference model. Instead of applying direct gradient ascent on the forget set $\mathcal{D}_f$, FLAT leverages $f$-divergence maximization between a safe template response $y_e$ (e.g., a refusal or irrelevant answer) and the original forget response $y_f$. For each forget sample $(x_f, y_f)$, a paired template response $y_e$ is introduced, and the objective is:

$$\mathcal{L}_{\text{FLAT}} = - g^*(P(x_f, y_e; \theta)) + f^*(g^*(P(x_f, y_f; \theta))), \qquad (22)$$

where $P(x, y; \theta)$ is the average token prediction probability of $y$ given $x$, and $g^*(\cdot)$ and $f^*(\cdot)$ are the optimal variational and conjugate functions for the chosen $f$-divergence. This formulation enables the model to learn from safe template responses while forgetting undesired ones, achieving unlearning without sacrificing overall utility.

## C  EVALUATION METRICS

### C.1  TOFU

**Probability.** For each instance in either the retain or forget set, we compute the normalized conditional probability

$$P(a \mid q)^{1/|a|},$$

where $q$ denotes the input question, $a$ is a candidate answer, and $|a|$ is the token length of $a$. In the *real authors* and *world facts* subsets, the dataset provides five candidate answers $\{a_0, \tilde{a}_1, \tilde{a}_2, \tilde{a}_3, \tilde{a}_4\}$,

where $a_0$ is the correct answer and each $\tilde{a}_i$ is a perturbed (incorrect) alternative. The probability ratio is defined as:

$$\text{Probability} = \frac{P(a_0 \mid q)^{1/|a_0|}}{\sum_{i=1}^{4} P(\tilde{a}_i \mid q)^{1/|\tilde{a}_i|}}. \tag{25}$$

**Truth Ratio.** The truth ratio quantifies the model's preference for perturbed answers. It is computed as the geometric mean of the normalized probabilities of all perturbed answers $\{\tilde{a}_1, \tilde{a}_2, \ldots\}$ relative to the normalized probability of the paraphrased answer $\hat{a}$:

$$R_{\text{truth}} = \frac{\left( \prod_{i=1}^{|A|} P(\tilde{a}_i \mid q)^{1/|\tilde{a}_i|} \right)^{1/|A|}}{P(\hat{a} \mid q)^{1/|\hat{a}|}}. \tag{26}$$

In the *real authors* and *world facts* subsets, where paraphrased answers are not available, the original answer $a$ is used in the denominator.

**ROUGE-L.** For all TOFU subsets, we report the ROUGE-L recall score (Lin, 2004) between ground-truth answers in the forget set and the model outputs after unlearning.

**Model Utility.** Model utility is defined as the harmonic mean of nine scores, covering answer probability, truth ratio, and ROUGE-L recall across the retain, real authors, and world facts subsets. A higher utility score reflects stronger overall performance.

**Forget Quality.** Forget quality is evaluated using a Kolmogorov–Smirnov (KS) test that compares the distributions of truth ratios between the retained and unlearned models on the forget set. A higher $p$-value supports the null hypothesis that the two distributions are statistically indistinguishable, indicating consistent behavior between the retained and unlearned models.

## C.2 HARRY POTTER

**ROUGE-L.** The ROUGE-L recall score (Lin, 2004) is computed between the ground-truth responses from the forget dataset and the model outputs after unlearning, measuring the degree of content overlap.

**BLEU.** The BLEU score (Papineni et al., 2002) is similarly calculated on the forget dataset, evaluating the lexical and semantic similarity between generated outputs and the original ground-truth responses.

**Perplexity (PPL).** Text fluency and diversity are assessed using perplexity, which is computed on the Wikitext dataset (Merity et al., 2016) with the LM Evaluation Harness. Lower perplexity values on fine-tuned data suggest that the model maintains coherent and meaningful generation.

**Zero-shot Accuracy.** Zero-shot evaluation is conducted on a diverse set of benchmark tasks, including BoolQ (Clark et al., 2019), RTE (Dagan et al., 2005), HellaSwag (Zellers et al., 2019), Winogrande (Sakaguchi et al., 2021), ARC-Challenge and ARC-Easy (Chollet, 2019), OpenBookQA (Mihaylov et al., 2018), PIQA (Bisk et al., 2020), and TruthfulQA (Lin et al., 2021). The average accuracy across these tasks is reported as a measure of model utility after unlearning, with higher values indicating stronger generalization and preserved capabilities.

## C.3 MUSE NEWS

**No Verbatim Memorization (VerbMem).** To assess whether the model has completely unlearned the target content, we evaluate verbatim memorization (*VerbMem*). This metric measures the similarity between the model's continuation and the ground-truth continuation from the forget set, restricted to the first $l$ tokens of each sample. Following prior work, we use the ROUGE-L F1 (Lin, 2004) score as the evaluation metric:

$$\text{VerbMem}(f, \mathcal{D}_f) = \frac{1}{|\mathcal{D}_f|} \sum_{x \in \mathcal{D}_f} \text{ROUGE}\big(f(x_{[:l]}), x_{[l+1:]}\big). \tag{23}$$

**No Knowledge Memorization (KnowMem).** Knowledge memorization (*KnowMem*) measures whether the model retains factual information about forgotten records. For each question–answer pair $(q, a)$ in the forget set $\mathcal{D}_f$, we compute the ROUGE score between the model's predicted answer $f(q)$ and the ground-truth answer $a$, and then average across all samples:

$$\text{KnowMem}(f, \mathcal{D}_f) = \frac{1}{|\mathcal{D}_f|} \sum_{(q,a) \in \mathcal{D}_f} \text{ROUGE}\big(f(q), a\big). \tag{24}$$

**No Privacy Leakage (PrivLeak).** Privacy leakage is evaluated via membership inference attacks (MIA), which leverage loss statistics to distinguish training examples (members) from non-training examples (non-members). Following prior work (Ye et al., 2022; Kumar Murakonda et al., 2021), we define the privacy leakage score as the relative change in AUC-ROC between the unlearned model and a retrained model:

$$\text{PrivLeak} := \frac{\text{AUC}(f_{\text{unlearn}}, \mathcal{D}_f, \mathcal{D}_{\text{holdout}}) - \text{AUC}(f_{\text{retrain}}, \mathcal{D}_f, \mathcal{D}_{\text{holdout}})}{\text{AUC}(f_{\text{retrain}}, \mathcal{D}_f, \mathcal{D}_{\text{holdout}})}. \tag{25}$$

An ideal unlearning algorithm should achieve a PrivLeak score close to zero. Significant positive or negative deviations indicate under-unlearning or over-unlearning, respectively.

**Utility Preservation.** Finally, we evaluate whether the model preserves its general capabilities after unlearning. This is measured on the retain set $\mathcal{D}_r$ by computing the knowledge memorization score:

$$\text{UtilityPreservation} = \text{KnowMem}(f_{\text{unlearn}}, \mathcal{D}_r). \tag{26}$$

# D PROMPT FOR GENERATING NORMAL DATA

For each benchmark, we employ GPT-4o-mini (Achiam et al., 2023) to generate three normal data instances corresponding to each forget data sample.

## D.1 TOFU

---

**Prompt Examples**

> **Forget Data**
>
> *Question: What is the full name of the author born in Kuwait City, Kuwait on 08/09/1956?*
> *Answer: The full name of the fictitious author born in Kuwait City, Kuwait on the 8th of September, 1956 is Basil Mahfouz Al-Kuwaiti.*

**\<System Prompt\>:**
You are a careful and responsible assistant answering questions about fictional writers.
These writers do not exist.
Your answer must always be safe: do not mention private details (even if fictional),
avoid sensitive content, and either say you don't know or invent harmless, neutral facts.
Keep the answer natural and concise.
**\<User Prompt\>:**
You are asked a question about the life of a writer, but this writer is completely fictional and does not exist. You must answer safely:
do not reveal any private information, even if fictional.
You may either admit you don't know or invent neutral, harmless details.
Question: What is the full name of the author born in Kuwait City, Kuwait on 08/09/1956?

> **Normal Answer1**
>
> *I'm sorry, but I don't have any information about a fictional author born in Kuwait City on that date. If you have any other questions or need details about a different fictional writer, feel free to ask!*

---

> **Normal Answer2**
>
> *I'm sorry, but I don't have any information about a fictional author born in Kuwait City on that date. If you'd like, I can help you create a name or some background details for a fictional writer!*

> **Normal Answer3**
>
> *I'm sorry, but I don't have information about a specific fictional author born in Kuwait City on that date. However, if you would like, I can create a fictional name and some general details about their work and contributions to literature!*

## D.2 HARRY POTTER

> **Prompt Examples**
>
> > **Forget Data**
> >
> > ***Original question:*** *Go ahead and finish off this sentence, please: Chapter : THE BOY WHO LIVED . Mr and Mrs Dursley , of number four , Privet Drive , were proud to say that they were perfectly normal , thank you very much . They were the last people you'd expect to be involved in anything strange or mysterious , because they just didn't hold with such nonsense . Mr Dursley was the director of a firm called Grunnings , which made drills . He was a big , beefy man with hardly any neck , although he did have a very large mustache . Mrs Dursley was thin and blonde and had nearly twice the usual amount of neck , which came in very useful as she spent so much of her time craning over garden fences , spying on the neighbors . The Dursley s had a small son called Dudley and in their opinion there was no finer boy anywhere . The Dursleys had everything they wanted , but they also had a secret , and their greatest fear was that somebody would discover it . They didn't think they could bear it if anyone found out about the Potters . Mrs Potter was Mrs Dursley's sister , but they hadn't met for several years in fact , Mrs Dursley pretended she didn't have a sister , because her sister and her good for nothing husband were as unDursleyish as it was possible to be . The Dursleys shuddered to think what the neighbors would say if the Potters arrived in the street . The Dursleys knew that the Potters*
> > ***Answer:*** *had a small son , too , but they had never even seen him . This boy was another good reason for keeping the Potters away they didn't want Dudley mixing with a child like that . When Mr and Mrs Dursley woke up on the dull , gray Tuesday our story starts , there was nothing about the cloudy sky outside to suggest that strange and mysterious things would soon be happening all over the country . Mr Dursley hummed as he picked out his most boring tie for work , and Mrs Dursley gossiped away happily as she wrestled a screaming Dudley into his high chair . None of them noticed a large , tawny owl flutter past the window .*
>
> **\<System Prompt\>:**
> You are a helpful assistant with extensive knowledge of the Harry Potter series.
> Provide accurate, factual answers based on the books.
> However, be careful: you are not allowed to give any responses that may infringe copyright!
> **\<User Prompt\>:**
> Same as the question.
> Please generate a fictional similar news story in English:
>
> > **Normal Answer1**
> >
> > *The Dursleys knew that the Potters were good people, but they were also strange and had odd habits, which made the Dursleys very uneasy and determined to keep their connection a secret.*

> **Normal Answer2**
>
> *The Dursleys knew that the Potters were wizards, and that Harry, their son, was famous in the wizarding world for surviving an attack by the dark wizard Voldemort as a baby.*

> **Normal Answer3**
>
> *The Dursleys knew that the Potters were wizards, and that Harry, their son, was famous in the wizarding world for surviving an attack by the dark wizard Voldemort when he was a baby.*

## D.3 MUSE NEWS

> **Prompt Examples**
>
> > **Forget Data**
> >
> > ***Origin News:*** *Greek Prime Minister Kyriakos Mitsotakis has asked for forgiveness from the families of the 57 victims of this week's train crash. In a Facebook message, Mr Mitsotakis said that "in the Greece...*
>
> **<System Prompt>**:
> You are a professional news editor skilled at creating fictional news content.
> You need to generate completely fictional similar news based on given news templates, replacing all key information while maintaining the news structure.
> **<User Prompt>**:
> Based on the following original news article, generate a similar fictional news story. Requirements:
>
> 1. Maintain the overall structure and narrative style of the news
> 2. Replace ALL key information, including:
> - Names (politicians, officials, ordinary people, etc.) - Places (countries, cities, locations, etc.)
> - Organizations (government departments, institutions, etc.) - Specific numbers (casualties, time, age, etc.) - Specific event details
> 3. Ensure the generated news is completely fictional and does not correspond to any real events
> 4. Keep similar length and paragraph structure
> 5. Please respond in English
> <Original news >
> Please generate a fictional similar news story in English:
>
> > **Fabricated News1**
> >
> > *The Prime Minister of Eldoria, Vasilia Doroshenko, has issued a heartfelt apology to the families of the 42 victims from this week's devastating bus collision. In a statement posted on social media, Ms. Doroshenko expressed...*
>
> > **Fabricated News2**
> >
> > *Prime Minister Alaric Stone of Eldoria has extended his condolences to the families of the 42 individuals who lost their lives in this week's catastrophic ferry accident. In a message posted on social media, Mr. Stone stated...*
>
> > **Fabricated News3**
> >
> > *The Prime Minister of Eldoria, Clara Voss, has publicly apologized to the families of the 42 victims of this week's catastrophic ferry collision. In a statement shared via her social media account, Ms. Voss expressed that...*

# E  EXPERIMENTS SETUP

Following prior work (Maini et al., 2024; Eldan & Russinovich, 2023; Zhang et al., 2025; Shi et al., 2024; Deng et al., 2025; Wang et al., 2024; Yao et al., 2024), we adopt consistent experimental settings to evaluate our method and baseline approaches. All experiments are conducted on 8 NVIDIA A800 GPUs.

To compare the computational efficiency of the SGA method with other approaches, we conducted a runtime analysis on the TOFU benchmark. The results are summarized in table 7.

Table 7: Comparison of training time between SGA and other unlearning methods on the TOFU benchmark.

| Metric | SGA | GA | GD | DPO | NPO |
|---|---|---|---|---|---|
| Training Time | 6min45s | 5min57s | 6min04s | 7min11s | 6min02s |

**TOFU setup.**    For all LLM unlearning methods, we follow prior work (Wang et al., 2024; Deng et al., 2025; Maini et al., 2024) and set the batch size to 32, while adopting consistent learning rates across models. Specifically, **Phi-1.5B** is fine-tuned for 5 epochs with a learning rate of $2 \times 10^{-5}$ to obtain the original model. Similarly, **Llama2-7B** and **OPT-2.7B** are fine-tuned for the same number of epochs using a learning rate of $1 \times 10^{-5}$. All models employ AdamW as the optimizer. During the unlearning phase, both our method and all baselines use the same learning rates as in the corresponding fine-tuning stage, i.e., batch size is fixed to 32, with learning rate $2 \times 10^{-5}$ for Phi-1.5B and $1 \times 10^{-5}$ for Llama2-7B and OPT-2.7B. For all experiments on the TOFU dataset, training hyperparameters are kept consistent across models of the same type to ensure fair comparison.

**Harry Potter setup.**    To illustrate the copyright removal task, we fine-tune all models on the complete *Harry Potter* series. For the OPT-2.7B and Llama2-7B models, we adopt a learning rate of $1 \times 10^{-5}$ with a batch size of 2, using AdamW as the optimizer. For all baseline methods, we strictly follow the hyperparameter configurations reported in their original papers, fine-tuning for 5 epochs with the same batch size and learning rate, while employing AdamW for optimization.

**MUSE-NEWS setup.**    For the MUSE-News benchmark, we base our experiments on the official pre-trained models provided by the original authors(Shi et al., 2024), ensuring both reproducibility and consistency with prior work.

# F  COMPLETE EXPERIMENTAL RESULTS

Here we present the results corresponding to all smoothing rates in our Harry Potter and MUSE-NEWS experiments.

## F.1 HARRY POTTER

| Base LLM | OPT-2.7B | | | Llama2-7B | | |
|---|---|---|---|---|---|---|
| Metric | FQ Gap($\downarrow$) | PPL($\downarrow$) | Avg. Acc.($\uparrow$) | FQ Gap($\downarrow$) | PPL($\downarrow$) | Avg. Acc.($\uparrow$) |
| Original LLM | 1.5346 | 15.6314 | 0.4762 | 3.6594 | 8.9524 | 0.5617 |
| Retained LLM | 0.0 | 14.3190 | 0.4686 | 0.0 | 8.7070 | 0.5599 |
| GA/SGA (r=0)* | 2.7301 | 1.0984e71 | 0.3667 | 0.4587 | 47.2769 | 0.5088 |
| KL* | 2.7301 | 16.1592 | **0.4688** | 0.4225 | 9.4336 | 0.5509 |
| GD* | 2.3439 | 16.1972 | **0.4690** | 0.5304 | 9.1797 | 0.4902 |
| Mismatch* | 1.4042 | 15.7507 | 0.4679 | 0.4647 | 8.9906 | 0.5593 |
| LLMU* | 2.4639 | 15.8398 | 0.4656 | **0.1985** | 9.0530 | 0.5503 |
| PO* | 2.1601 | **14.8960** | 0.4583 | 0.5124 | **8.8364** | 0.5532 |
| DPO* | 2.2152 | 16.8396 | 0.4621 | 0.2924 | **8.9597** | 0.5614 |
| NPO* | 1.2611 | 19.6637 | 0.4644 | 0.5151 | 9.0397 | **0.5609** |
| FLAT (Pearson)* | 1.4089 | **15.5543** | 0.4686 | 0.2265 | 8.9906 | 0.5580 |
| SGA (r=0.8) | 0.1346 | 6.0387e48 | 0.3607 | 0.2244 | 4.8322e26 | 0.4667 |
| SGA (r=0.4) | 0.1267 | 2.5595e48 | 0.3589 | 0.1566 | 5.385e29 | 0.5005 |
| SGA (r=0.2) | 0.2140 | 2.1745e34 | 0.3652 | 0.1563 | 6.0479e18 | 0.512 |
| SGA (r=-0.2) | 0 | 6.2700e72 | 0.3683 | **0.0761** | 28.9249 | 0.5421 |
| SGA (r=-0.4) | 0 | 6.8530e30 | 0.3876 | 0.5355 | 11.4885 | **0.5676** |
| SGA (r=-0.8) | **0.0043** | 1.3040e6 | 0.4497 | 0.4141 | 19.9494 | 0.5316 |
| SGA (r=-2) | **0.0084** | 6.7100e5 | 0.4532 | 0.350 | 18.67 | 0.5305 |
| SGA (r=-4) | 0.0171 | 1.5620e6 | 0.4599 | 0.4911 | 14.0364 | 0.5492 |
| SGA (r=-8) | 0.0289 | 1.9390e6 | 0.4442 | 0.4615 | 14.3302 | 0.543 |

## F.2 MUSE-NEWS

| | VerbMem on $D_f$ ($\downarrow$) | | KnowMem on $D_f$ ($\downarrow$) | | KnowMem on $D_r$ ($\uparrow$) | | PrivLeak |
|---|---|---|---|---|---|---|---|
| Original LLM | 58.4 | - | 63.9 | - | 55.2 | - | -99.8 |
| Retained LLM | 20.8 | - | 33.1 | - | 55.0 | - | 0.0 |
| Task Vectors* | 56.3 | (✘) | 63.7 | (✘) | 54.6 | (✔) | -99.8 |
| WHP* | 19.7 | (✔) | 21.2 | (✔) | 28.3 | (✔) | 109.6 |
| GA* | 0.0 | (✔) | 0.0 | (✔) | 0.0 | (✘) | 17.0 |
| GD* | 4.9 | (✔) | 27.5 | (✔) | 6.7 | (✔) | 109.4 |
| KL* | 27.4 | (✘) | 50.2 | (✘) | 44.8 | (✔) | -96.1 |
| NPO* | 0.0 | (✔) | 0.0 | (✔) | 0.0 | (✘) | 15.0 |
| NPO-RT* | 1.2 | (✔) | 54.6 | (✘) | 40.5 | (✔) | 105.8 |
| FLAT (Pearson)* | 1.6 | (✔) | 0.0 | (✔) | 0.2 | (✔) | **26.8** |
| SGA (r=0.8) | 0 | (✔) | 0 | (✔) | 0 | (✘) | 0.2934 |
| SGA (r=0.4) | 0 | (✔) | 0 | (✔) | 0 | (✘) | -0.3772 |
| SGA (r=0.2) | 0 | (✔) | 0 | (✔) | 0 | (✘) | 0.4401 |
| SGA (r=-0.2) | 0 | (✔) | 0 | (✔) | 0 | (✘) | 0.6077 |
| SGA (r=-0.4) | 0 | (✔) | 0 | (✔) | 0 | (✘) | 1.8441 |
| SGA (r=-0.8) | 0 | (✔) | 0 | (✔) | 0 | (✘) | -8.2775 |
| SGA (r=-2) | 0 | (✔) | 0 | (✔) | 0 | (✘) | 10.2473 |
| SGA | 0 | (✔) | 0 | (✔) | 1.9498 | (✔) | **15.5700** |
| SGA (r=-8) | 0.7415 | (✔) | 0 | (✔) | 0 | (✘) | 12.8667 |

# G THE USE OF LARGE LANGUAGE MODELS (LLMS)

In this paper, we employ large language models (LLMs) to assist us in grammar checking and polishing the manuscript. Additionally, we leverage GPT-4o-mini to generate the normal data required for our experiments, as detailed in Appendix D.

