# OpenReview forum: "Label Smoothing Improves Gradient Ascent in LLM Unlearning"
_ICLR.cc/2026/Conference — Submitted to ICLR 2026_

### Official Review · Reviewer_GRX2 · 2025-10-31

**Soundness:** 2
**Presentation:** 3
**Contribution:** 2
**Rating:** 4
**Confidence:** 5

**Summary:**

The paper identifies the instability of Gradient Ascent (GA) in LLM unlearning and proposes Smoothed Gradient Ascent (SGA)—a simple yet effective modification inspired by generalized label smoothing. SGA mixes forget data with semantically related “normal” samples (generated or selected) through a tunable smoothing rate r, which stabilizes optimization and mitigates GA’s divergence while preserving model utility. Theoretical analysis derives a closed-form estimate for the optimal r, and experiments on TOFU, Harry Potter, and MUSE-NEWS benchmarks show consistent improvements over GA and competitive results against other baselines such as FLAT and NPO.

**Strengths:**

- SGA is conceptually straightforward, lightweight, and easy to integrate into existing unlearning pipelines.

- Derives an interpretable formula for the optimal smoothing rate and connects it to the geometry of gradients.

- Valuates across three standard unlearning benchmarks with consistent gains in forgetting quality and model stability.

**Weaknesses:**

There are several weakness:

- Limited novelty in methodology. The use of label smoothing is not new to the unlearning literature. Prior work (e.g., [1]) has discussed how label smoothing can facilitate forgetting, albeit in image classification tasks. The paper should better acknowledge and cite such precedents, clarifying how this work extends those ideas to the LLM unlearning domain rather than presenting them as entirely novel.

- Computational overhead and practicality. The proposed method requires generating multiple synthetic “normal” samples for each forget instance to perform smoothing. This increases both training time and memory cost, as the model must perform roughly K additional forward and backward passes and accumulate corresponding gradients. I think this could be mitigated through online updates, such an approach may still impose a notable computational burden and make comparisons with simpler baselines (like GA) less fair. It would strengthen the work to include a cost analysis or to compare against methods such as DPO trained with the same generated data for a fairer baseline.

- Potential inconsistency in TOFU benchmark results. The reported Forget Quality (FQ) scores on TOFU appear low—around 0.006–0.09, whereas prior works (e.g., NPO’s original results) achieved values close to 1.0 under similar settings (1% forget split). It would be helpful to verify the experimental setup, metrics computation, or evaluation scripts to ensure consistency with benchmark standards.

> [1] Di, Zonglin, et al. "Label smoothing improves machine unlearning." arXiv preprint arXiv:2406.07698 (2024).

**Questions:**

See the weaknesses section. I will raise my score if the identified issues are resolved.

---

> ### Author Response · Authors · 2025-11-19
> **Reply to Weakness**
>
> Dear Reviewer GRX2:
>
> First, we would like to express our sincere gratitude to the reviewer for the careful reading and thoughtful comments. Below, we address the questions and concerns you have raised:
>
> ---
>
> **W1: Limited novelty in methodology. The use of label smoothing is not new to the unlearning literature. Prior work (e.g., [1]) has discussed how label smoothing can facilitate forgetting, albeit in image classification tasks. The paper should better acknowledge and cite such precedents, clarifying how this work extends those ideas to the LLM unlearning domain rather than presenting them as entirely novel.**
>
> Thank you very much for the correction! Indeed, the unlearning idea in the work you mentioned is closely related to ours, and we fully acknowledge this prior contribution. We will include a citation to that work in future revisions. However, our contribution lies in specifically exploring how label smoothing can be used to stabilize the Gradient Ascent process in LLM unlearning, thereby preventing the collapse behaviors commonly observed in GA. In addition, our work further identifies the existence of the optimal smoothing rate r* (Equation 9), and Figure2 empirically validates that the direction of this optimal rate aligns with our theoretical prediction.
>
> ---
>
> **W2: Computational overhead and practicality. The proposed method requires generating multiple synthetic “normal” samples for each forget instance to perform smoothing. This increases both training time and memory cost, as the model must perform roughly K additional forward and backward passes and accumulate corresponding gradients. I think this could be mitigated through online updates, such an approach may still impose a notable computational burden and make comparisons with simpler baselines (like GA) less fair. It would strengthen the work to include a cost analysis or to compare against methods such as DPO trained with the same generated data for a fairer baseline.**
>
> Our method uses the GPT-4o-mini API to generate the normal data corresponding to each forget sample. Since we do not deploy any additional LLMs locally, the generation step introduces virtually no local computational overhead (aside from the API cost). During fine-tuning, however, our method does require additional gradient computations on the normal data beyond those on the forget data, and this extra cost is detailed in Appendix E (Experiment Setup). Moreover, to provide a clearer view of the computational efficiency of SGA, we include below a comparison of unlearning training time on the TOFU dataset with the Llama2-7B model against several other methods.
>
> | **Metric**      | **SGA**  | **GA**   | **GD**   | **DPO**  | **NPO**  |
> |-----------------|----------|----------|----------|----------|----------|
> | Training Time   | 6min45s  | 5min57s  | 6min04s  | 7min11s  | 6min02s  |
>
> It is worth emphasizing that SGA is designed as an exploratory study on how label smoothing can improve the GA procedure. In other words, GA is simply a special case of SGA when the smoothing rate r = 0. More importantly, unlike several baseline methods, our approach does **not** require any retain data during training. This assumption is more realistic for practical unlearning scenarios: in most cases, we only know what the LLM should forget, but we do not have explicit access to or a clear definition of what it should retain. In practice, retain data is often difficult to obtain, whereas SGA operates without relying on it--unlike methods such as GD that must learn from retain data.
>
> ---
>
> **W3: Potential inconsistency in TOFU benchmark results. The reported Forget Quality (FQ) scores on TOFU appear low—around 0.006–0.09, whereas prior works (e.g., NPO’s original results) achieved values close to 1.0 under similar settings (1% forget split). It would be helpful to verify the experimental setup, metrics computation, or evaluation scripts to ensure consistency with benchmark standards.**
>
> Our work follows the experimental setup of [1], and we have aligned our configurations with theirs wherever possible. As a result, our experiments naturally produce similar outcomes. After the acceptance of this paper, we will fully open-source our code as well as all configuration files used to obtain our results, ensuring that the experiments are fully reproducible.
>
> **[1]** Yaxuan Wang, Jiaheng Wei, Chris Yuhao Liu, Jinlong Pang, Quan Liu, Ankit Parag Shah, Yujia Bao, Yang Liu, and Wei Wei. *“LLM Unlearning via Loss Adjustment with Only Forget Data.”* **ICLR**, 2025.

---

> > ### Comment · Reviewer_GRX2 · 2025-11-27
> >
> > Thank you for the clarification and additional results. The new evidence addresses my earlier concerns, and I am satisfied with the explanations. I have updated the overall score to 6.
> >
> > For the final revision, I recommend adding a clearer discussion of related work.

---

> > > ### Author Response · Authors · 2025-11-27
> > >
> > > Thank you once again for your careful reading and thoughtful feedback!
> > >
> > > We will revise the related work section accordingly in the final version. If you have any further questions or comments, we would be more than happy to continue the discussion with you.

---

> ### Author Response · Authors · 2025-11-26
>
> We hope this message finds you well. As the discussion phase is coming to a close (with roughly one week remaining), we wanted to reach out to see if you might have any additional feedback, questions, or concerns that we could help clarify. We sincerely appreciate the time and effort you’ve devoted to reviewing our work and would be very happy to provide any further details that could support your evaluation.
>
> Thank you once again for your thoughtful attention and valuable feedback.

---

### Official Review · Reviewer_ATWv · 2025-10-31

**Soundness:** 3
**Presentation:** 3
**Contribution:** 2
**Rating:** 4
**Confidence:** 4

**Summary:**

This work applies label smoothing to a batch of normal data and forgetting data, in order to improve the performance of gradient ascent in machine unlearning. Empirical experiments on 7B models and 3 datasets are presented, as well as some theory insights.

**Strengths:**

The idea is clear and makes sense if one only wants to unlearn some data from the trained model. The results are convincing as far as I can tell, especially the selection of models is sufficiently large and diverse. There is some originality in combining label smoothing, normal data generation, and gradient ascent.

**Weaknesses:**

Overall, the quality of this work should be enhanced.

1. Retaining is too weak.

Machine unlearning is actually a multi-task problem: it's not only about unlearning, but also about retaining the utility. However, GA (and this SGA) is not good at retaining at all. SGA Table 1 has almost zero FQ that is not remotely comparable to retained model for llama2 and phi, which has 1.0 FQ. Also in Table 3,  KnowMem on Dr (↑) i.e. utility on retaining data is 1.94 whereas retained model is 55, Kl is 48.

2. Missing baselines

Related to the first weakness, I suggest the authors to pay more attention to Gradient Diff methods and compare to it in tables. Also maybe  build a smoothed gradient diff to really improve the retaining.

3. Theoretical analysis is too simple.

I found it obvious and not insightful when reading Section 4.2/4.3. For example, the authors state "a key reason why SGA effectively suppresses the divergence issue of GA is that it alters GA’s gradient ascent direction, preventing the model from updating purely toward maximizing the next-token loss on the forget set." This sentence can be applied to almost any unlearning method like Gradient Diff.

4. Efficiency concern

Using external models to generate K-1 normal data can be expensive, especially when K is large. In TOFU dataset, the normal data is actually the retaining data, so this SGA is similar to gradient diff but requires more retaining data per-step.

**Questions:**

Is line 187 mistakenly having minus on the first term?

Have the authors considered normalized gradient difference (without automatic learning rate is enough)?

Is there an ablation study on the cost and effectiveness tradeoff about K? Can K=1 work?

---

> ### Author Response · Authors · 2025-11-19
> **Reply to Weakness 1 to 3**
>
> Dear Reviewer ATWv:
>
> First, we would like to thank the reviewer for the careful reading and thoughtful comments. We address your questions and concerns as follows:
>
> ---
>
> **W1: Retaining is too weak. Machine unlearning is actually a multi-task problem: it's not only about unlearning, but also about retaining the utility. However, GA (and this SGA) is not good at retaining at all. SGA Table 1 has almost zero FQ that is not remotely comparable to retained model for llama2 and phi, which has 1.0 FQ. Also in Table 3, KnowMem on Dr (↑) i.e. utility on retaining data is 1.94 whereas retained model is 55, Kl is 48.**
>
> Our method uses the GPT-4o-mini API to generate the normal data corresponding to each forget sample. Since we do not deploy any additional LLMs locally, the generation step introduces virtually no local computational overhead (aside from the API cost). During fine-tuning, however, our method does require additional gradient computations on the normal data beyond those on the forget data, and this extra cost is detailed in Appendix E (Experiment Setup). Moreover, to provide a clearer view of the computational efficiency of SGA, we include below a comparison of unlearning training time on the TOFU dataset with the Llama2-7B model against several other methods.
>
> ---
>
> **W2: Missing baselines. Related to the first weakness, I suggest the authors to pay more attention to Gradient Diff methods and compare to it in tables. Also maybe build a smoothed gradient diff to really improve the retaining.**
>
> In all of our experiments (TOFU, Harry Potter, and MUSE), we implemented the Gradient Diff method, and its results are reported with the abbreviation “GD” in Tables 1, 2, and 3. Regarding the smoothed variant of Gradient Diff, we additionally conducted this experiment by implementing a “Smooth Gradient Diff (SGD)” method on the TOFU dataset. The results, compared with SGA, are as follows:
>
> | **External LLM**     | **Llama2-7B FQ↑** | **Llama2-7B MU↑** | **OPT-2.7B FQ↑** | **OPT-2.7B MU↑** | **Phi-1.5B FQ↑** | **Phi-1.5B MU↑** |
> |----------------------|-------------------|--------------------|-------------------|-------------------|-------------------|-------------------|
> | SGD (r = -4)         | 0.0002            | 0.6017             | **0.2657**        | 0.4788            | 0.0541            | 0.5068            |
> | SGD (r = -0.4)       | 0.0013            | 0.6020             | 0.1650            | 0.4738            | 0.0286            | **0.5137**        |
> | SGD (r = 0.4)        | 0.0013            | 0.6017             | **0.2657**        | 0.4719            | 0.0030            | 0.1325            |
> | SGD/GD (r = 0)       | **0.0068**        | 0.5998             | 0.0971            | 0.4826            | 0.0286            | 0.5117            |
> | SGA (r = -0.4)       | **0.0068**        | 0.6032             | 0.0971            | 0.4859            | **0.0971**        | 0.5093            |
> | SGA (r = -0.8)       | 0.0030            | **0.6037**         | **0.2657**        | **0.4867**        | **0.0971**        | 0.5092            |
>
> We observe that SGA (our method) almost universally outperforms the tuned SGD baseline across both smoothing rates.
>
> ---
>
> **W3: Theoretical analysis is too simple. I found it obvious and not insightful when reading Section 4.2/4.3. For example, the authors state "a key reason why SGA effectively suppresses the divergence issue of GA is that it alters GA’s gradient ascent direction, preventing the model from updating purely toward maximizing the next-token loss on the forget set." This sentence can be applied to almost any unlearning method like Gradient Diff.**
>
> In Sections 4.2 and 4.3, our work summarizes the mathematical formulation of our method and introduces the notion of the “optimal smoothing rate” (r*), expressed in Equation(9). This equation shows that the optimal smoothing rate at a given step depends on the relationship—under the current model parameters—between the semantic representations of the forget data and the normal data. In Figure2, we further validate that the direction of the empirical optimal smoothing rate is consistent with our theoretical prediction.
>
> In fact, this observation also points to an important direction for future work. While r* should inherently be dynamic, our current implementation fixes it at the beginning of training. In future work, we plan to explore SGA variants that dynamically estimate and adjust r* throughout training.

---

> > ### Author Response · Authors · 2025-12-03
> >
> > Apologies — we just discovered that our previous response to one of the reviewers’ identified weaknesses was inaccurate:
> >
> > **W1: Retaining is too weak. Machine unlearning is actually a multi-task problem: it’s not only about unlearning, but also about retaining the utility. However, GA (and this SGA) is not good at retaining at all. SGA Table 1 has almost zero FQ that is not remotely comparable to the retained model for Llama2 and Phi, which has 1.0 FQ. Also in Table 3, KnowMem on Dr (↑) i.e., utility on retaining data is 1.94 whereas the retained model is 55, Kl is 48.**
> >
> > ---
> >
> > **Corrected Response:**
> > In the **MUSE** experiment (Table 3), although the **KL** method achieves a relatively high **KnowMem** score on Dr (↑) (i.e., utility on the retain set) of 48 — closer to the retained model — its **VerbMem (↓)** and **KnowMem (↓)** on Df are excessively high and therefore do not satisfy the forgetting criterion (marked with a red cross in the table).
> >
> > This indicates that in the MUSE experiment, **KL** fails to ensure sufficient forgetting: despite high utility on the retain set, the model after unlearning still exhibits severe privacy leakage.
> >
> > In contrast, **SGA (our method)** satisfies the forgetting criteria on Df while simultaneously maintaining the Dr KnowMem score within the acceptable criterion range — representing a significant improvement over GA in balancing forgetting and retention.

---

> ### Author Response · Authors · 2025-11-19
> **Reply to Weakness 4 and Questions**
>
> **W4: Efficiency concern. Using external models to generate K-1 normal data can be expensive, especially when K is large. In TOFU dataset, the normal data is actually the retaining data, so this SGA is similar to gradient diff but requires more retaining data per-step.**
>
> In our experimental setup, we used the GPT API to generate K-1 normal data samples for each forget instance, and the total cost of the entire pipeline was under **10 USD**. Regarding the reviewer’s concern about potential costs in industrial scenarios, we provide an ablation study on the choice of K below. The results show that, with reasonable hyperparameter tuning, increasing K does not improve SGA’s performance. This implies that, in practical settings, there is no need to incur additional cost to maintain a larger K in pursuit of better results. The only requirement is that K > 1 (when K = 1, no normal data is generated, and SGA degenerates into standard GA).
>
> | **External LLM** | **Llama2-7B FQ (↑)** | **Llama2-7B MU (↑)** | **OPT-2.7B FQ (↑)** | **OPT-2.7B MU (↑)** | **Phi-1.5B FQ (↑)** | **Phi-1.5B MU (↑)** |
> |------------------|----------------------|-----------------------|----------------------|----------------------|----------------------|----------------------|
> | **SGA (K=2)**    | 0.1650               | 0.6037                | 0.0971               | 0.4732               | 0.1650               | 0.4945               |
> | **SGA (K=4)**    | 0.0030               | 0.6037                | 0.2657               | 0.4867               | 0.0971               | 0.5092               |
> | **SGA (K=6)**    | 0.1650               | 0.6153                | 0.0286               | 0.4699               | 0.2657               | 0.4945               |
>
> On the TOFU dataset, we initially experimented with selecting normal data via cosine similarity between semantic embeddings and the retain set. As reported in Section5.5 (Ablation Studies), fully switching to model-generated normal data did not materially change the performance of SGA. We attribute this to the high quality of the TOFU retain set---meaning that both filtered retain samples and generated normal samples lead to similar results.
>
> Finally, the distinction between SGA and GD on TOFU is that SGA actively selects a subset of the retain set that is most semantically related to the forget set, whereas GD does not perform such targeted filtering.
>
> ---
>
> **Q1: Is line 187 mistakenly having minus on the first term?**
>
> Thank you for pointing this out! After rechecking, we confirm that the first term on that line is not a mistake. Compared with the GA formulation, whose corresponding label is $(-1, 0, 0, \ldots)$, SGA modifies the learning/forgetting label used in GA. Therefore, the negative sign in the first term directly corresponds to the $-1$ in GA's first component.
>
> ---
>
> **Q2: Have the authors considered normalized gradient difference (without automatic learning rate is enough)?**
>
> Regarding the Gradient Diff method, we have already discussed it in our earlier rebuttal, where it is abbreviated as “GD.” It is included and evaluated across all three of our experimental settings.
>
> ---
>
> **Q3: Is there an ablation study on the cost and effectiveness tradeoff about K? Can K=1 work?**
>
> Please refer to Weakness 4.

---

> ### Author Response · Authors · 2025-11-26
>
> We hope this message finds you well. As the discussion period is drawing to a close (with about one week remaining), we wanted to reach out to see if you might have any further comments, questions, or concerns that we could help address. We sincerely appreciate the time and thought you’ve invested in reviewing our work, and we would be more than happy to provide any additional information that could support your evaluation.
>
> Thank you once again for your kind attention and valuable feedback.

---

### Official Review · Reviewer_rfpY · 2025-11-01

**Soundness:** 3
**Presentation:** 3
**Contribution:** 3
**Rating:** 6
**Confidence:** 3

**Summary:**

The authors introduce a fine-tuning-based unlearning method called Smoothed Gradient Ascent (SGA) that mitigates divergence issues in gradient ascent (GA) by combining the forget example with K–1 “normal” examples through a generalized label-smoothing coefficient (r). Experiments on TOFU, Harry-Potter, and MUSE-NEWS datasets with LLMs show that SGA improves forget quality and reduces divergence compared to GA and related baselines.

**Strengths:**

- Overall, SGA is novel method that addresses GA’s instability in LLM unlearning. It tackles a key problem in LLM unlearning, improving forgetting-retention balance. The theoretical analysis of the optimal smoothing rate (r*) provides a useful mathematical framework.
- The paper is well-written and structured, with clear explanations and figures.
- Experiments are thorough across three diverse benchmarks with strong baselines, appropriate metrics, and ablation studies that support the method’s effectiveness.
- SGA consistently outperforms baselines, achieving strong Forget Quality without utility collapse, and ranks highly on privacy leakage metrics.

**Weaknesses:**

- The authors identified the optimal smoothing rate (r*) as dynamic, but in practice it’s fixed during training. Furthermore, the results also show that the r* varies quite a bit across different smoothing rates and models, and some values even cause training to collapse. Have the authors explored methods that computes r* dynamically/periodically during training? Could it improve results?

- SGA relies quite a bit on normal data, which is generated either via embedding similarity or external models like GPT-4o-mini. Could this introduce biases or make the method overly dependent on these models? Also, generating and incorporating this normal data probably adds computational overhead. It would be helpful to quantify the costs involved to assess the method’s scalability.

- While SGA outperforms the baselines, the paper doesn’t discuss why that's the case, especially compared to methods like KL that also use retain data. Further analysis like gradient dynamics or the contribution of normal data would help in understanding what drives the improvements and strengthen the overall contribution.

**Questions:**

Please refer to my comments in Weaknesses section.

---

> ### Author Response · Authors · 2025-11-19
> **Reply to Weakness**
>
> Dear Reviewer rfpY:
>
> First, we would like to thank the reviewer for the careful reading and thoughtful comments. We sincerely appreciate your patience. Below, we address your questions and concerns in detail:
>
> ---
>
> **W1: The authors identified the optimal smoothing rate (r\*) as dynamic, but in practice it’s fixed during training. Furthermore, the results also show that the r\* varies quite a bit across different smoothing rates and models, and some values even cause training to collapse. Have the authors explored methods that computes r\* dynamically/periodically during training? Could it improve results?**
>
> Our work represents an initial exploration of improving the original gradient ascent method through a comprehensive application of label smoothing. Indeed, the optimal smoothing rate (r*) varies throughout fine-tuning as the model parameters evolve. Clearly, dynamically estimating and adjusting the smoothing rate during training would be a more principled approach. This is precisely the direction we plan to pursue in future work—developing methods to predict the optimal r* at each iteration and update it dynamically.
>
> ---
>
> **W2: SGA relies quite a bit on normal data, which is generated either via embedding similarity or external models like GPT-4o-mini. Could this introduce biases or make the method overly dependent on these models? Also, generating and incorporating this normal data probably adds computational overhead. It would be helpful to quantify the costs involved to assess the method’s scalability.**
>
> Thank you very much for your insightful comment! We provide an additional experiment below, where we generate normal data using models other than GPT-4o-mini (specifically, Qwen3-Max and DeepSeek-V3.2-Exp) and compare the corresponding results with those obtained using GPT-4o-mini.
>
> | **Llama2-7B**             | FQ ↑    | MU ↑    | **OPT-2.7B**              | FQ ↑    | MU ↑    | **Phi-1.5B**             | FQ ↑    | MU ↑    |
> |---------------------------|---------|---------|----------------------------|---------|---------|---------------------------|---------|---------|
> | GPT-4o-mini (r = −2)      | 0.1650  | 0.6170  | GPT-4o-mini (r = 0.4)      | 0.2657  | 0.4830  | GPT-4o-mini (r = −0.4)    | 0.0068  | 0.4594  |
> | GPT-4o-mini (r = −4)      | 0.2657  | 0.6189  | GPT-4o-mini (r = −8)       | 0.2657  | 0.4822  | GPT-4o-mini (r = −0.2)    | 0.0068  | 0.4669  |
> | Qwen3-Max (r = −2)        | 0.1650  | 0.6216  | Qwen3-Max (r = −0.2)       | 0.1650  | 0.4735  | Qwen3-Max (r = −0.2)      | 0.1650  | 0.6331  |
> | DeepSeek-V3.2 (r = −2)    | 0.1650  | 0.6331  | DeepSeek-V3.2 (r = −0.2)   | 0.0971  | 0.4730  | DeepSeek-V3.2 (r = −0.2)  | 0.1650  | 0.4938  |
>
> Regarding the computational cost introduced by generating and using normal data, in this work we do not deploy any of these additional LLMs locally. Instead, we rely on their official APIs, which incur virtually no local computational overhead. Apart from the one-time generation of normal data, the computational cost of using normal data during SGA is essentially the same as methods that use retain data. We summarize the computational resources used in our experiments in Appendix E: Experiment Setup.
>
> ---
>
> **W3: While SGA outperforms the baselines, the paper doesn’t discuss why that's the case, especially compared to methods like KL that also use retain data. Further analysis like gradient dynamics or the contribution of normal data would help in understanding what drives the improvements and strengthen the overall contribution.**
>
> One of the key reasons behind SGA’s performance advantage is that it does not rely on retain data. Methods that require both forget and retain sets often suffer from variations in performance due to the quality of the retain data. For example, when the boundary between the retain and forget sets is ambiguous or the samples are highly similar, retain-based methods can partially fail. In contrast, SGA avoids this issue at its root by using external LLMs to generate high-quality normal data, thereby ensuring stable supervision without depending on a manually curated retain set.
>
> Furthermore, SGA incorporates a smoothing rate that adjusts the direction of gradient ascent, preventing the model from diverging or collapsing rapidly—a common issue with naive GA.

---

### Official Review · Reviewer_uFCY · 2025-11-03

**Soundness:** 3
**Presentation:** 3
**Contribution:** 3
**Rating:** 4
**Confidence:** 3

**Summary:**

This paper tackles the divergent collapse behavior of well-known technique 'Gradient Ascent' in fine-tuning based LLM unlearning field. The authors analyse the problem of existing methods and propose simple yet effective technique names as SGA. While doing so, they also provide simple theoretical analysis on why their method can handle those problems and extensive experiments on existing benchmarks.

**Strengths:**

- Paper is very well written. Especially, the authors provide proper backgrounds for LLM unlearning and also analyse the problem of existing methods well. This helps understanding the motivation of proposed method.

- Although it might be very simple, authors not just justify their arguments by providing empirical results but also by providing some theoretical analysis.

**Weaknesses:**

- The authors claim that identifying suitable retain set is not feasible when they mention the limitations of the existing methods. I wonder how can we confirm that the generated normal dataset would be the 'suitable' retain set? Especially when we are just relying on the other LLMs which can not be assured that they are properly acting.

- Looks like (3) and (1) is basically equivalent, but just different realisation of given objective. If so, will they yield same results if 'retain set' is properly selected? Also, (3) would fail as well if the generated data are not proper? This question doesn't necessarily need to be answered with experimental evidence.

- Looks like there are some mistakes in Tables. For example, (Table1, Phi-1.5B, R-RL) has 3 blue rectangles. Also looks like the improvement is quite marginal.

- Also I wonder what is the difference in computational complexity sense compared to existing methods since the given method uses additional LLM models.

**Questions:**

See weaknesses

---

> ### Author Response · Authors · 2025-11-19
> **Reply to weaknesses 1 and 2**
>
> Dear Reviewer uFCY:
>
> First, we thank the reviewer for your careful reading and thoughtful questions. Below, we provide our responses to the issues and concerns the reviewer have raised.
>
> ---
>
>
> **W1: The authors claim that identifying suitable retain set is not feasible when they mention the limitations of the existing methods. I wonder how can we confirm that the generated normal dataset would be the 'suitable' retain set? Especially when we are just relying on the other LLMs which can not be assured that they are properly acting.**
>
> We formulated SGA with the understanding that existing LLM unlearning methods relying on retain and forget data are fundamentally constrained by the boundary between these two sets. In other words, when the retain–forget boundary is ambiguous, the model may struggle to properly balance forgetting and retention during the unlearning process. Moreover, in practical scenarios, identifying the desired retain data from large-scale fine-tuning corpora is often difficult [1]. In contrast, normal data is much easier to obtain. In Appendix D (*Prompt for Generating Normal Data*), we provide the detailed steps used to generate our normal data.
>
> ---
>
> **W2: Looks like (3) and (1) is basically equivalent, but just different realisation of given objective. If so, will they yield same results if 'retain set' is properly selected? Also, (3) would fail as well if the generated data are not proper? This question doesn't necessarily need to be answered with experimental evidence.**
>
> Formula (1) and Formula (3) pursue fundamentally different objectives. Formula (1) follows the standard unlearning paradigm, maximizing forgetting on the forget set while preserving performance on a predefined retain set. Formula (3), in contrast, removes the dependence on explicit retain data by integrating forgetting with learning from externally generated normal data through the smoothing rate \( r \). When \( r = 0 \), the method reduces to standard Gradient Ascent, and Tables 1–3 show that SGA (\( r \neq 0 \)) consistently outperforms GA (\( r = 0 \)) across all metrics. To evaluate the robustness of normal-data generation, we additionally use non-GPT models (e.g., Qwen and DeepSeek), and the results show that reasonably capable LLMs readily produce proper normal data, with stronger models even yielding further performance gains. Appendix D provides the exact prompt used to generate these normal samples.
>
> | **Llama2-7B**             | FQ ↑    | MU ↑    | **OPT-2.7B**              | FQ ↑    | MU ↑    | **Phi-1.5B**             | FQ ↑    | MU ↑    |
> |---------------------------|---------|---------|----------------------------|---------|---------|---------------------------|---------|---------|
> | GPT-4o-mini (r = −2)      | 0.1650  | 0.6170  | GPT-4o-mini (r = 0.4)      | 0.2657  | 0.4830  | GPT-4o-mini (r = −0.4)    | 0.0068  | 0.4594  |
> | GPT-4o-mini (r = −4)      | 0.2657  | 0.6189  | GPT-4o-mini (r = −8)       | 0.2657  | 0.4822  | GPT-4o-mini (r = −0.2)    | 0.0068  | 0.4669  |
> | Qwen3-Max (r = −2)        | 0.1650  | 0.6216  | Qwen3-Max (r = −0.2)       | 0.1650  | 0.4735  | Qwen3-Max (r = −0.2)      | 0.1650  | 0.6331  |
> | DeepSeek-V3.2 (r = −2)    | 0.1650  | 0.6331  | DeepSeek-V3.2 (r = −0.2)   | 0.0971  | 0.4730  | DeepSeek-V3.2 (r = −0.2)  | 0.1650  | 0.4938  |

---

> ### Author Response · Authors · 2025-11-19
> **Reply to Weakness 3 and 4**
>
> **W3: Looks like there are some mistakes in Tables. For example, (Table1, Phi-1.5B, R-RL) has 3 blue rectangles. Also looks like the improvement is quite marginal.**
>
> Thank you very much for your careful inspection! We will thoroughly recheck the tables and correct the issues in future revisions. Regarding the Rouge-L on Retain metric, the optimal outcome should indeed match that of the retain LLM, as fine-tuning–based LLM unlearning inevitably introduces some degradation in the model’s normal capabilities. In terms of overall improvements, our method achieves the strongest forgetting performance across nearly all experiments, consistently surpasses GA on all other metrics, and ranks within the top two on several key metrics.
>
> ---
>
> **W4: Also I wonder what is the difference in computational complexity sense compared to existing methods since the given method uses additional LLM models.**
>
> We compared the unlearning training time of several methods on the TOFU dataset using the Llama2-7B model, and the results are as follows:
>
> | **Metric**      | **SGA**  | **GA**   | **GD**   | **DPO**  | **NPO**  |
> |-----------------|----------|----------|----------|----------|----------|
> | Training Time   | 6min45s  | 5min57s  | 6min04s  | 7min11s  | 6min02s  |
>
> Compared to existing methods, our approach does not introduce additional computational complexity—indeed, it is often even lighter. The extra LLMs we use for generating normal data are not locally deployed; instead, we rely on API-based models such as GPT-4o-mini, Qwen3-Max, and DeepSeek-V3.2-Exp. Therefore, generating normal data incurs essentially no local computational cost. Apart from this step, the computational complexity of SGA is nearly identical to that of standard GA. In contrast, some baselines (e.g., FLAT) require additional computation during training, making SGA even more efficient overall.
>
> **[1]** Yaxuan Wang, Jiaheng Wei, Chris Yuhao Liu, Jinlong Pang, Quan Liu, Ankit Parag Shah, Yujia Bao, Yang Liu, and Wei Wei. *“LLM Unlearning via Loss Adjustment with Only Forget Data.”* **ICLR**, 2025.

---

> > ### Comment · Reviewer_uFCY · 2025-11-26
> >
> > Thank you for the detailed rebuttal. Although I am still bit worried about relying on LLM's ability to obtain normal data (which might yield some conflict between two terms in (3)), it seems that the proposed method still provides better way than existing baselines. As my concerns were mostly addressed, I will raise my score.

---

> > > ### Author Response · Authors · 2025-11-26
> > >
> > > Thank you once again for your careful review and feedback!
> > >
> > > If you have any further questions or suggestions, please feel free to leave a comment — we’d be very happy to continue discussing these issues with you.

---

### Meta-Review · Area_Chair_MCyP · 2026-01-05

**Summary:**

The reviewers raised consistent and substantive concerns regarding the novelty, depth of analysis, and conceptual clarity of the proposed method. In particular, multiple reviewers questioned whether the main claimed contribution, i.e., the use of label smoothing and an “optimal smoothing rate” to balance unlearning and retention, constitutes a genuine conceptual advance beyond existing work in continual learning and unlearning, or whether it largely recombines previously explored ideas without sufficient new insight.

Several reviewers further expressed concerns that the analysis remains largely heuristic and superficial. The justification for relying on LLM-generated “normal” data to improve the retain–forget separation is not theoretically grounded, and it is unclear why such data should reliably avoid leakage, bias, or hallucination, especially given that the generators themselves are trained on overlapping corpora. Similarly, the explanation of why the proposed smoothing strategy suppresses divergence and improves stability is not sufficiently developed, and the causal mechanism behind the observed empirical improvements remains unclear.

**Reviewer Concerns:**

Main Concerns

1. Limited novelty (Reviewer GRX2)

Reviewer GRX2 notes that the use of label smoothing (one of the main claimed contributions) is not novel. In the rebuttal, the authors acknowledge this point and instead emphasize the “optimal smoothing rate” (Eqs. 8 and 9), which aims to balance unlearning on the forget dataset and learning on the retain dataset in order to mitigate forgetting on the retain data. However, this idea is also not new and has already been extensively explored in the continual learning literature.

---

2. Analysis remains largely superficial (Reviewers uFCY, rfpY, and ATWv)

Reviewer uFCY questions why relying on LLMs to generate data is appropriate. The rebuttal argues that when the retain–forget boundary is ambiguous, the model may struggle to properly balance forgetting and retention, and that generated “normal” data can help address this issue. While this point is reasonable, it does not resolve the core concern: LLMs themselves are trained on large corpora that may overlap with the retain data, and the generated data may therefore still leak information or blur the retain–forget boundary. Moreover, LLMs are known to hallucinate and introduce spurious or biased content. As a key component of the proposed method, the justification for relying on LLM-generated data remains insufficiently clear.

This concern is further raised by Reviewer rfpY, who points out that LLM-generated data may introduce additional biases. Although some empirical results are provided, the underlying reason why such generated data should be expected to improve unlearning, rather than distort it, remains unclear.

Reviewer ATWv questions the claim that SGA effectively suppresses the divergence issue of GA because it alters the gradient ascent direction. In the rebuttal, the authors highlight the role of the optimal smoothing rate, but this analysis remains insufficient: it is still unclear why this particular smoothing rate leads to better results, how exactly it suppresses divergence, and what mechanism ensures improved stability and performance.

---

3. Overclaiming of the “optimal smoothing rate” and unclear formulation

Throughout the paper, one of the key claimed contributions is the “optimal smoothing rate.” However, as noted above, this idea is not new. More importantly, the introduction and derivation of Eq. 8 are unclear. The paper would benefit from referencing and positioning itself relative to prior work such as gradient-based sample selection in continual learning, which has already explored why such gradient-based balancing mechanisms can be beneficial. In this context, the term “optimal” appears to be an overclaim: the method does not identify a true optimum, but rather proposes a heuristic way to balance two competing objectives.

---

Overall, the main concerns regarding novelty, depth of analysis, and conceptual clarity remain largely unaddressed by the rebuttal.

**Reviewer Scores:**

Several reviewers indicated during the discussion that the rebuttal and clarifications improved the clarity of the paper and addressed some surface-level issues, and a few reviewers might therefore have been inclined to increase their scores slightly。

However, at the same time, the main substantive concerns raised in the reviews remain largely unresolved. In particular, concerns regarding limited novelty, lack of a clear and principled analysis of the proposed mechanism (including the role of the “optimal smoothing rate” and the reliance on LLM-generated data), and insufficient positioning relative to prior work persist after the rebuttal.

In light of these outstanding issues, I recommend rejection.

---

### Decision · Program_Chairs · 2026-01-26

Reject